



# *Digital Carbonate Rock Physics*

Erik H. Saenger[1,2], Stephanie Vialle[3], Maxim Lebedev[3], David Uribe[2], Maria Osorno[4], Mandy Duda[1], and Holger Steeb[4].

[1]International Geothermal Centre, Bochum, 44801, Germany
[2]Ruhr-University, Bochum, 44801, Germany
[3]Curtin University, Perth, Australia
[4]University of Stuttgart, 70569 Stuttgart, Germany

*Correspondence to*: E. H. Saenger (erik.saenger@rub.de)

Keywords: Carbonates; Effective elastic properties; Permeability; Digital Rock Physics; Nano-indentation;

**Abstract.** Modern estimation of rock properties combines imaging with advanced numerical simulations, an approach known as Digital Rock Physics (DRP). In this paper we suggest a specific
segmentation procedure of X-Ray micro-Computed Tomography data with two different resolutions for two sets of carbonate rock samples. These carbonates were already characterized in detail in a previous laboratory study which we complement with nano-indentation experiments. In a first step a non-local mean filter is applied to the raw image data. We then apply different thresholds to identify pores and solid phases. Because of a non-neglectable amount of unresolved micro-porosity ("micritic phase") we
also define intermediate phases. Based on this segmentation we determine porosity-dependent values for P- and S-wave velocities as well as for the intrinsic permeability. The porosity measured in the laboratory is then used to predict the effective rock properties for a comparison with experimental data. Anisotropy is observed for some sub-samples, but seems to be insignificant in our case. Because of the complexity of carbonates we suggest to use DRP as a complementary tool for rock characterization in
addition to classical experimental methods.





## 1 Introduction

Three-dimensional information on rock microstructures is important for a better understanding of physical phenomena and for rock characterization on the micro-scale. Various destructive and non-destructive methods for obtaining a 3D image of the rock microstructure exist (Madonna et al., 2013, Cnudde and Boone, 2013, Fusseis et al., 2014, and references therein). The most common non-destructive 3D imaging technique for rock samples is X-Ray Computed Tomography (XRCT). A common problem, however, is a clear trade-off between sample size and resolution. For each material a specific, and large enough, sample size is required to ensure that the selected volume is representative of the physical property to be computed. It can, however, be at the expense of a lost of pore features resolution. In the last decade, the X-Ray micro-Computed Tomography (micro-XRCT) method became widely available and many modern studies have made use of it to obtain 3D rock images. The resolution of micro-XRCT of up to $(0.6\mu m)^3$ (voxel size) is high enough to image the spatial distribution of grains, pores, and pore fluids.

Rock images from 3D XRCT can be used for predicting properties such as porosity, permeability, pore size distribution, effective elastic moduli, or electrical conductivity. For example, Darcy permeability can be predicted by numerically simulating single-phase fluid flow through 3D rock models, with the numerical results being in reasonable agreement with laboratory measurements (e.g. Saenger et al., 2015). In this case, the resolution of the micro-XRCT technique is sufficient, because fluid pathways predominantly follow larger pores. However, if the porosity is much smaller than 1μm the agreement might be less satisfactory due to resolution limitations. On the other hand, mechanical properties, such as the effective elastic moduli, strongly depend on the microstructural details of the rock, which stay unresolved by the micro-XRCT technique. The inability to fully characterize the microstructural details of a rock sample can lead to a disagreement between numerical estimates of mechanical properties based on micro-XRCT images and laboratory data.

An example of such a disagreement between laboratory and Digital Rock Physics (DRP) estimates is described in Andrä et al. (2013a,b). In these benchmark papers a comparison between different numerical methods is presented. All DRP estimates of the effective elastic bulk modulus use the same segmented dataset. Regardless of the approach all numerical predictions overestimate the bulk modulus




measured in the laboratory. This conclusion is mostly based on Berea sandstone although carbonates are considered in this study as well. However, also Jouini et al. (2015) reports about an overestimation of effective elastic properties of carbonates by DRP. Therefore we conclude here that the digital rock images themselves and/or the computational workflow has to be improved to provide better estimate of

effective properties of rocks. In this paper we consider in detail a carbonate dataset and suggest techniques to achieve a better agreement between numerical predictions and laboratory measurements. Our study is complementary to the DRP-carbonate studies performed in Lopez et al. (2012), Andrä et al. (2013a,b) and Jouini et al. (2015).

## 2 Rock Samples and Laboratory Characterization

### 2.1 Carbonate samples

We studied samples of two carbonates from the Upper Cretaceous carbonate system of the Gargano-Murge region (Southern Italy). *Carb-A* is a limestone from the Paleocene-Eocene Peschici formation, and *Carb-B* is a micritic mudstone from the Late Cretaceous Monte Acuto formation (Martinis and Pavan, 1967; Cremonini et al., 1971). Both carbonates are composed of nearly 100% calcite (Scotallero

et al., 2014). SEM images showing the microstructure of these two samples are given in Figure 1 of Vialle et al (2013). Both carbonate samples display a matrix of micrite (abbreviation for *micr*ocrystalline calc*ite*) whose grain size is typically 1–4μm (Moshier, 1989), but the texture of these micrites is different between the two samples. Following the classification of micrite microtexture proposed by Lambert et al. (2006) and Deville de Periere et al. (2011), the micrite in sample *Carb-A* is

mainly a "tight micrite", spatially varying from anhedral compact to fused, with grains typically 1–2μm in diameter; micrite in sample *Carb-B* is a "porous micrite" varying from rounded to subrounded, with anhedral to subhedral, rounded grains, typically 2–4μm in diameter. Beside a micrite matrix, sample *Carb-A* exhibits vuggy-like pores either rounded, up to about 60μm, or more elongated, up to 300μm in length. Sample *Carb-B* exhibits a spar calcite cement of grains typically 10 to 100's of micrometers in

diameter, as well as rounded vugs up to about 100–200μm.

He-grain density, bulk density and resulting porosity, as well as air-permeability, were previously


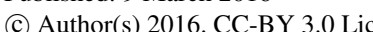


measured at room pressure and temperature on core plugs 1'' (2.5 cm) in length and diameter (Vialle et al., 2013). The associated errors did not exceed 0.5%, 1%, and 2%, respectively. P- and S-wave velocities, at 1 and 0.7 MHz frequency, respectively, were acquired on dry samples under increasing (up to 30 MPa) and decreasing hydrostatic stress. Velocities were measured by using a pulse-transmission

technique. The errors in $V_P$ and $V_S$ are about 1%. The results are listed in Tables 1 and 2. The obtained grain densities ($2.69 \pm 0.01 \text{g/cm}^3$ and $2.70 \pm 0.01 \text{g/cm}^3$, for samples *Carb-A* and *Carb-B*, respectively) are in agreement with a mineralogy of pure calcite (Mavko et al., 2009).

**2.2 Nano-indentation**

Nanoindentation tests were performed to obtain stiffness (Young's modulus) of the carbonates at the

micrometer-scale. These tests were performed on room-dried samples consisting of two small irregular pieces, about 5 mm thick and with a surface of a few $\text{cm}^2$, taken from the cuttings of the 1'' core plugs of *Carb-A* and *Carb-B*. Prior to testing, the surface of each sample was polished with carbide paper (grit 120). Roughness (Sq) of the surface measured by DS 95 AFM system (Semilab) on 10μm x 10μm areas was 1.4μm for *Carb-A* and not measured for *Carb-B* (RMS values). The IBIS nano-indentation system

(Model B, Fisher-Crips Laboratories Pty.Ltd.) is equipped with a Berkovich-type diamond indenter (Lebedev et al, 2014) and was used in a static mode: the tests consist in continuously recording the load, *P,* and the displacement, *h,* of the indenter as it pushes into and withdraws from the surface of the sample. A constant maximum loading force of 10mN and an initial contact force of 0.15mN were used. In total, 961 (31 x 31) measurements were performed on a 300 x 300μm surface with a spacing of 10μm

between measurement points.

Typically, the extraction of the mechanical properties is achieved by using the *P-h* curves and by applying a continuum scale mechanical model to obtain the indentation modulus *M*:

$$M \overset{def}{=} \frac{\sqrt{\pi}}{2} \frac{S}{\sqrt{A_c}} \qquad (1)$$





where $S$ the is unloading indentation stiffness $S = \left( dP/dh \right)_{h=h_{max}}$ and $A_c$ the contact area, extrapolated from the maximum penetration depth $h_{max}$ and using the relation $A_c = 24.5 . h_{max}^2$ according to the geometry of Berkovitch-type indenters.

Data was further corrected considering deviation of the indenter tip from ideal geometry, initial penetration into the rock below a load threshold and compliance of the loading column, leading to a nominal uncertainty of indentation moduli of $< 2$GPa.

Young's moduli , $E$, can be calculated from the indentation moduli according to

$$\frac{1}{M} = \frac{1-v^2}{E} + \frac{1-v_i^2}{E_i} \quad (2)$$

Indenter properties are $E_i = 1220$GPa and $v_i = 0.06$, according to Klein & Cardinale (1992) and Fischer-Cripps, (2004) for diamond material. Each performed measurement covers a projected surface of about 40µm² on average (which corresponds to an equilateral triangle with a side of 6µm), and contains both pores and solid grains. A Poisson's ratio, $v$, has also to be assumed for each nano-indentation measurement: even though we can compute it from the laboratory ultrasonic P- and S- wave measurements at the core scale, there is no reason to assume that this value is constant for each individual measurement. Figure 1 displays the distribution of the indentation moduli for both *Carb-A* (left) and *Carb-B* (right).

## 3 The CT-datasets

### 3.1 Procedure to get CT-Images

Two samples were prepared for imaging with micro-XRCT from the cuttings, one from *Carb-A* and one from *Carb-B*. A cylindrical-shape sample of 1.5cm in height and 2mm in diameter was achieved by gently grinding the cuttings, first on the side on a rock saw blade, and then by hand using sand paper (grit 120). This procedure allows obtaining very thin cylinders while minimizing mechanical damage that classical drilling would produce. These cylindrical samples were then glued with Crystalbond509 (SPI suppliers) on a 2mm diameter flat-head metal pin, which was itself inserted in the core holder of the micro-tomograph. The 3D X-ray Microscope Versa XRM 500 (Zeiss - XRadia) was used with a X-

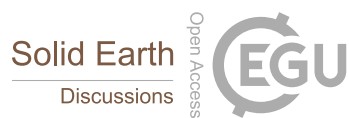

ray energy of 60keV. Two different settings of source-to-sample distance were used to achieve two nominal voxel sizes of $(3.4\mu m)^3$ and $(0.6\mu m)^3$ for *Carb-A*, and of $(3.4\mu m)^3$ ("low" resolution) and $(1.1\mu m)^3$ ("high" resolution) for *Carb-B*. X-ray microscope and image acquisition settings are summarized in Table 3 for each of the two samples.

5 The number of radiographic projections acquired during sample imaging with "low" and "high" resolutions were 3001 and 5001, respectively. The total scanning time for one sample was about 8h. Initial cone-beam 3D image reconstruction was performed using internal "Reconstructor" software (Zeiss-XRadia). To illuminate geometrical artefacts during reconstruction a secondary reference was acquired for samples image with maximum resolution.

10 **3.2 Segmentation procedure**

In addition to solid grains and pore space different micritic phases are visible in the raw images of the scans entailing an advanced segmentation procedure. For our segmentation (Figure 2) we select a region of interest (ROI) from the raw data of the two types of carbonates with two different resolutions (Tab 3). The ROI is subdivided in eight partly overlapping sub-volumes, each of a size of $400^3$ voxels 15 (Figures 3 and 4). For the low-resolution images, it gives thus eight cubes with a side of 1.37mm for both samples, and for the low-resolution images, 8 cubes with a side of 0.25mm for *Carb-A* and of 0.46mm for *Carb-B*.

Our segmentation workflow is applied to the full ROI including all subvolumes. Image enhancement and segmentation steps were carried out using the software package Avizo Fire 9 (FEI Visualization 20 Sciences Group). Before actual segmentation the image noise and scan artefacts are reduced while preserving interfaces using a non-local mean image filter calibrated for the appropriate dimensions and kernel window sizes. Note that every step of image enhancement changes the original data set affecting subsequent steps required for data analysis.

The image-enhanced datasets were segmented into classes using global thresholds for the covered range 25 of grey-values. Considering that the samples represent quasi monomineralic calcitic rocks we identified the following classes illustrated in Figures 5 and 6:

- high-confidence pores (illustrated with dark blue color),





- high-confidence mineral (illustrated with dark red color),

- and five classes in between the mineral class and the pore class.

A non-negligible part of the pore space is below the resolution limit of the µ-CT scans. cf. results of mercury intrusion, Figure 6 of Vialle et al., 2013, showing pores and pore throats as low as 0.06µm.

## 4 Numerical Results

In order to numerically calculate the effective intrinsic permeability $k$ of the digitized rock sample we calculate the fluxes under creeping flow condition based on a parallelized Stokes-solver. The parallelized Finite Differences-based Stokes-solver is suitable for the calculation of effective hydraulic parameters for low and high porous materials (cf. Osorno et al., 2015). Using volume averaging technique, we coarse-grain the local velocity field $\boldsymbol{u(x)}$ obtaining the global velocity component $u_m$ in flow direction.

The intrinsic permeability $k$ is calculated with Darcy's law:

$$k = \frac{\mu \, u_m}{\Delta p}. \qquad (3)$$

The pressure gradient $\Delta p$ is imposed with pressure boundary conditions in the numerical simulations. The dynamic viscosity of the pore fluid is $\mu$.

To obtain effective P- and S-wave velocities of the digitized rock samples we use a technique described in detail in Saenger et al. (2011) and references therein. The basic idea of this approach is to study speeds of elastic waves through heterogeneous materials in the long wavelength limit (pore size ≪ wavelength) using the RSG FD algorithm (Saenger et al., 2000) for the simulation of elastic wave velocities (cf. Andrä et al. 2013a,b).



## 4.1 High Resolution

### 4.1.1 Permeability

Permeability calculations were realized for selected sub-samples of the *Carb-A* and *Carb-B* samples. The domain size of the *Carb-A* (0.43 mm) and *Carb-B* (0.78 mm) high-resolution samples is smaller

than the low resolution (2.4 mm for *Carb-A* and *Carb-B*), i.e. less representative of the material, therefore we investigate numerically only the most relevant subsamples. The samples are selected in a way that their porosities are closest to the experimentally determined porosities. Further, to discuss extreme values, we additionally chose samples where the difference between numerical and experimental porosity is maximized.

Figure 7 left hand side displays the intrinsic permeability calculated for the *Carb-A* high resolution. The porosity range of the subsamples is higher than the experimentally determined porosity. In addition, the numerically calculated permeability values are significantly lower than the values obtained for the low-resolution samples. Figure 7 right hand side shows the results of the intrinsic permeability calculated for the selected high-resolution samples of the *Carb-B*. It can be observed, that the high-resolution sample

show a much lower variation between the extreme values.

From the results of the high-resolution samples, *Carb-A* and *Carb-B*, it could be observed that the variation in pores channels arrangement is significant and the permeability in the different subsamples of the same material does not necessarily increase with the porosity increment.

### 4.1.2 Elasticity

Several micritic phases have been identified in the raw images of the carbonate rock. The porosity of these regions cannot be determined exactly, as some pores are below the resolution of the scans: typically, micrites exhibit pore sizes with a maximum diameter of 1µm (Moshier, 1989; Cantrell and Hagerty, 1999), and pore sizes as low as 0.06µm have been measured by MICP for the samples under investigation (Vialle et al., 2013). To account for the unresolved pore space we perform a number of

two-phase simulations. For these simulations we assign vacuum properties to the pore phase, while the rest of the digital image including the micritic phase will be assumed to be solid with the mineral properties of calcite (e.g. Andrae et al., 2013b). For the second simulation, we assign vacuum properties





to the pores and the first micritic phase, while the rest will be assumed to be solid. We continue this way for all micritic phases, so that the last simulation assigns the mineral properties of calcite only to the high-confidence mineral phase. By this technique we obtain a porosity-velocity trend (Figure 8) for any random selection of high-resolution sub-samples for *Carb-A* and *Carb-B*:

$$vp = 6259.1 \text{m/s} - \phi * 9640 \text{m/s} + 3381 \text{m/s} * \phi^2 \qquad (4)$$

$$vs = 3237.2 \text{m/s} - \phi * 3237.2 \text{m/s} \qquad (5)$$

This porosity-velocity trend is exactly the same as has been observed for a carbonate dataset from a different location used in Saenger et al. (2014). In their paper this trend has been observed for three different resolutions (65nm, 1µm, 4µm).

## 4.2 Low Resolution

### 4.2.1 Permeability

Similar to the procedure of the numerical simulation for elasticity we vary the sample porosity. This way we get six possible domains for each subsample with different porosities. To reduce computational times for the Stokes flow simulation we eliminate the disconnected pores. Some sub-samples solid-pore configurations with the lower porosities do not present connected pores, and we assume for the effective permeability $k$=0.

Figure 9 present the permeability values for the *Carb-A* (left hand side) and *Carb-B* (right hand side) samples as a function of the porosity. Additionally we performed the Stokes flow simulations in three directions (X, Y and Z) for the *Carb-A* sample. Figure 11 (left hand side) shows that the *Carb-A* sample permeability is anisotropic with a variation between directions of up to 80%. In some sub-samples the permeability value varies by up to two orders of magnitude.

The largest sample available for the low resolution scan of *Carb-A* is 2.4mm x 2.4mm x 2.4mm ($689^3$ voxels). The permeability calculated for this sample is 13D for a porosity of 0.173.

### 4.2.2 Elasticity

For the low-resolution scans we repeat the two-phase simulations for *Carb-A* and *Carb-B* as described in section 4.1.2 (Figure 10). Interestingly, the two-phase trend given by Equations (4) and (5) is





confirmed in three cases: P- and S-wave velocities of *Carb-A* and S-wave velocities of *Carb-B*. For the case of *Carb-B* and P-wave velocities we observe a slightly different trend (cf. eq. (4)) that we illustrate with a blue dashed-dotted line in the left hand side of Figure 10:

$$vp = 6259.1 m/s - \phi * 7970 m/s + 1700 m/s * \phi^2 \qquad (6)$$

Especially in the low-resolution case we expect to have images with a large amount of unresolved porosity, mainly due to micritic phases. Therefore we perform multi-phase simulations and vary the porosity by assigning effective elastic properties to an interval of micritic phases (always starting with the class closest to the high-confidence pore phase). As effective medium approach we use the trend given by the simulations using two single phases only (Equations (4) and (5)). Despite the interval of

micritic phases we assign vacuum values for the high-confidence pores and use for the remaining phases the known elastic moduli of calcite (e.g. Andrae et al., 2013b). The results are displayed with green dots for *Carb-B* in Figure 10. We repeat the procedure with different intervals of the micritic phases. There are three interesting observations: (1) The resulting effective velocities are always significantly below the observed two-phase trend, (2), the curves for different intervals will intersect

each other, and (3), the experimental determined velocities for high confining pressures are between the multi-phase results and the two-phase trend as described above.

## 5 Discussion

In this paper we compare results from laboratory investigations with numerical estimates based on digital images. Note that in laboratory experiments we use samples on the cm-scale for the

determination of permeability and ultrasonic velocities and compare it with DRP-predictions based on images on the mm-scale. Especially because of the known heterogeneity of carbonates there is always a risk that the selected scanned area is not representative compared to the full sample size used for laboratory characterizations.

### 5.1 Discussion Experimental Characterization

Even with the highest resolution currently available in micro-XRCT imaging there will be a significant amount of unresolved pore features which need to be treated in the DRP workflow (Saenger et al.,



2015). On the grey-scale intensity level histograms of the low- and high-resolution images of the micro-CT scanning (Figure 5) this is reflected in a continuum in the intensity levels between the phase identified as pores and the phase identified as calcite grains. In this paper we have dealt with these micritic phases by replacing, step-by-step, and in a cumulative way, each of the micritic phases by void, and establishing a porosity-velocity trend. The nano-indentation technique provides a measure for the distribution of effective elastic properties at the micrometer-scale, and can thus potentially constrain the input parameters for the different phases identified during the segmentation. To be able to do so, nano-indentation needs to provide bulk and shear moduli from each of the measurements (load-displacement curves) and we need to obtain effective bulk and shear moduli values for each of the identified phases in the micro-tomography (pores, calcite grains and the five micritic phases). However, if nano-indentation technique is a well-established technique in material sciences, which deals with homogeneous, purely elastic materials, this is, as of today, not yet the case for rocks, which are heterogeneous materials with both elastic and non-elastic behaviour (creep). Though nano-indentation tests provide significant insights into elastic properties of heterogeneous rocks such as carbonates (Lebedev et al., 2014; Vialle & Lebedev, 2015) or shale (Ulm & Abousleiman, 2006; Abousleiman et al., 2007), there are still some points to be looked at before using the derived values of Young's (or shear and bulk) moduli in a quantitative way for DRP: value of Poisson's ratio to be used, effect of surface roughness, local mechanical damage induced on the sample's surface by polishing techniques, etc. Nonetheless, the histograms of the indentation moduli of both samples show a broad distribution of moduli values ranging from very low values (a few GPa, where the indenter tip measures stiffness of an area mostly made of a pore) to values consistent with calcite. The existence of these intermediate values is consistent with the existence of micritic phases identified with X-ray tomography. However, we did not observe two peaks in the histogram for the nano-indentation results (Figure 1) in contrast to the histograms of the scanned micro-XRCT-images (Figure 5). Therefore the direct translation of moduli derived from nano-indentation remains also to be difficult. The "resolution" of nano-indentation used in this study allows for determining effective elastic properties at slightly bigger scales than that used here for the micro-XRCT.


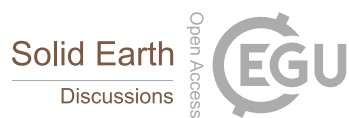

Regnet et al. (2014) showed that there is a relationship between micrite microstructure and laboratory ultrasonic velocities on core samples, with samples with higher content of 'tight micrite' exhibiting higher velocities and samples with higher content of 'microporous micrite' exhibiting lower velocities. Studied core samples were though a mixture of different types of micrite and the measured velocities

represent effective properties at the core scale. This observation is reflected in the established porosity-velocity trends (Equations 4 and 5): micritic phases with density closer to that of calcite ('tight micrites') have higher velocities than micritic phases with lower density closer to that of pores ("microporous micrites').

## 5.2 Discussion Porosity: Experiments vs. Digital Rock Physics

After the segmentation it is also possible to estimate the porosities of the samples. Based on the suggested workflow described in section 3.2 there will be a lower and upper bound. For the lower bound we will treat only the "high-confidence pores" as pores; for the upper bound we count only "high-confidence minerals" as minerals.

We obtain a porosity range between 25% and 35% for the high-resolution data of *Carb-A* and a range

between 7.5% and 31% for the low-resolution data. We observe that the mean value is in rough agreement with the experimentally determined porosity of 16.7% (Vialle et a. 2013) only for the low-resolution case. Although also the experimental value comes with an error we conclude that the high-resolution dataset for *Carb-A* is maybe not representative for the full sample. In case of *Carb-B* the intervals range from 13% to 45% and 7% to 48% for the high-resolution and low-resolution case,

respectively. Here the mean value is in both cases closer to the experimentally determined porosity of 29.4%.

We conclude that the porosity values of carbonates using micro-XRCT-data will only provide estimates with a relatively high uncertainty due to the significant amount of unresolved pore features in the images. An indication is the result of the mercury-intrusion experiments presented by Vialle et al.

(2013): The pore throats of the micritic phase are mainly below the resolution of available micro-XRCT devices.



### 5.3 Discussion Permeability: Experiments vs. Digital Rock Physics

Permeability numerically estimated for *Carb-B* presents an error of 97% on average with respect to the experimental value. In some cases the error is as low as 55%. The numerical error in comparison with the experimental values is within the expected range for the numerical method at these porosities, cf.

Table 1 in Osorno et al. (2015).

Experimental results for *Carb-A* sample are below the measurement error tolerance. This could imply a sample with no connected pores between the inlet and outlet defined for the experiment. The numerical estimation of the permeability for the *Carb-A* low resolution sample (on average 7 D) is four orders of magnitude higher than the experimental estimation. In the high resolution case the numerical estimation

is closer to experimental results (on average 100 mD), but the porosity presents a large numerical error, therefore we do not take this domain as representative of the sample. However the numerically calculated permeability does not differ much from values found in the literature for porous materials with alike porosity (cf. Andrä et al., 2013b). On the other hand it is observed that for a porosity below 25% permeability values of carbonates can span several orders of magnitudes (e.g. Figure 3 of Vialle et

al. 2013). Therefore we suggest considering a statistically significant number of samples to characterize a formation.

### 5.4 Discussion Elasticity: Experiments vs. Digital Rock Physics

There are two important observations. The two-phase trend (displayed with solid and dashed-dotted lines) seems to be an upper bound for velocities. This data-driven upper bound is much stricter than the

bound given by Hashin-Shtrikman (see Figure 8) and is now confirmed for several carbonates using several resolutions (this study and Saenger et al. 2014). Only for *Carb-B* we observe a slightly different trend for the low-resolution case for P-waves (equation (6) and Figure 10 right hand side).

The trend given by the envelope of the multi-phase simulations (displayed by green dots in Figure 10) is not a strict lower bound, because the shape will strongly depend on the applied method to determine

effective elastic properties for areas which are below the resolution limit of the used XRCT-technique. The best choice to our knowledge is the two-phase trend discussed above. This trend can be regarded as "carbonate-data-driven" effective medium approach. We suggest implementing here in the future also



the findings of the nano-indentation experiments. However, we observe that the velocities obtained for the multi-phase simulations are in a reasonable agreement with laboratory measurements. This is the case for a known porosity determined in complementary laboratory studies (see also section 5.1). For carbonates the distribution of the micritic phases and their effective elastic behavior is crucial to predict

the effective wave speeds.

## 5.5 Discussion Anisotropy: Elasticity vs. Permeability

In general we don't observe any significant anisotropy for permeability and for velocities of the considered samples. However, a few samples are out of this general trend. One example is a sub-sample of *Carb-A* (low-resolution case), for which we show the results for P-wave velocities and permeabilities

in Figure 11. Interestingly, the significant anisotropy for the permeability is not present for the velocity.

## 6 Summary

With the current imaging techniques it remains difficult to resolve microstructures (on sub-micrometer scale) and image a representative volume at the same time, which is essential to understand the effective material properties of rocks. The porosity of the rock samples is the most relevant parameter. To

overcome this problem, we have conducted a careful calibration of DRP estimates with laboratory data. For carbonate samples it is difficult to estimate the porosity from raw-CT data. Therefore, we use our presented numerical results in an inverse way. We suggest using the porosity determination from the laboratory and go back to our low-resolution results given in Figure 9 and 10. With a given porosity we can estimate the permeability and the effective wave velocities.

In case of the studied samples *Carb-A* and *Carb-B*, we can predict P- and S-wave velocities with a good agreement to laboratory results. The predicted permeability values are only in good agreement for *Carb-B*. Most probably the low-resolution image of *Carb-A* is not representative for the sample used in the laboratory.

However, for carbonate rock the resolution of the XRCT is not sufficient for a more exact estimate

because the micritic phases cannot be resolved. The effective elastic properties have to be approximated. Our suggestion is to use the trend of the two-phase simulations. The implemented





workflow in this paper can be applied in general for numerical estimates of mechanical and transport properties of carbonates. Because of the known strong heterogeneity of carbonates we suggest to use a statistically significant amount of digital images to characterize a formation.

## Acknowledgements

Erik H. Saenger would like to thank ExxonMobil for the support of some ideas presented in this study. This work was partially funded by Curtin Reservoir Geophysical Consortium (CRGC). The authors thank the National Geosequestration Laboratory (NGL) of Australia for providing access to the X-ray microscope VersaXRM-500 (Xradia - Zeiss Ltd) and to the Nanoindentation system (Fisher-Crips Laboratories Pty.Ltd.). Funding for this facility was provided by the Australian Federal Government.

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



**Figures**

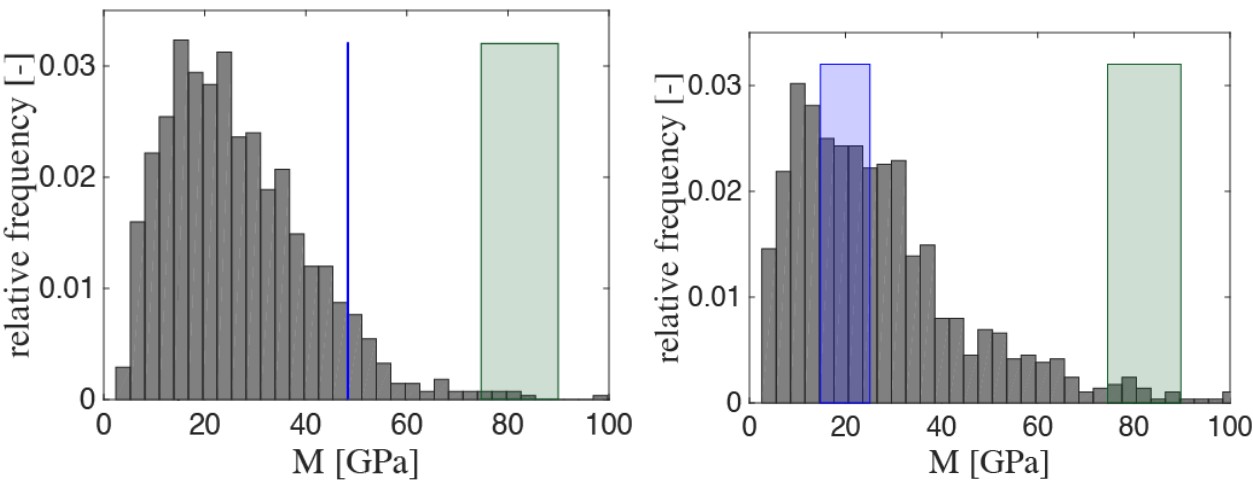

5    Fig. 1: Nano-Indentation results for *Carb-A* (left hand side) and *Carb-B* (right hand side). In blue we

illustrate the corresponding moduli range from ultrasonic experiments on dry samples from 0 to 30MPa

confining pressure, and in green we illustrate the moduli range given by the solid anisotropic calcite

crystal. Overall we observe that the medium effective indentation module M is slightly stiffer for *Carb-*

*A* (26GPa vs. 25GPa).

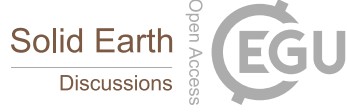



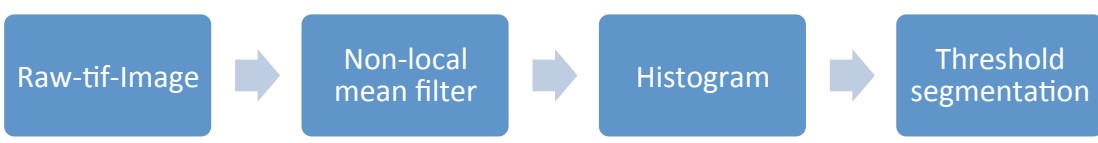

Fig. 2: Simplified segmentation workflow as applied in this study.

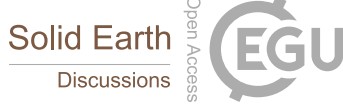

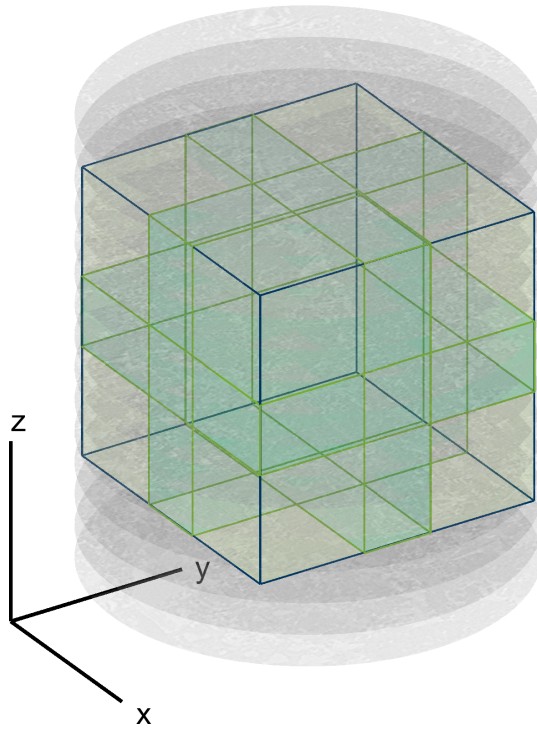

Fig. 3: Sketch to illustrate the segmentation geometry (here for *Carb-B*; high-resolution). The full cube is subdivided into eight partly intersecting sub-volumes with a size of $400^3$ gridpoints. Those sub-volumes are used in the numerical simulations to estimate effective material properties.



Fig. 4: Slices of the raw tiff-images of the scanned samples. Top row: *Carb-A* with high (left) and low-resolution (right). Bottom row: *Carb-B* with high (left) and low resolution (right).



5 Fig 5.: Color-coded histograms of the scanned samples. Top row: *Carb-A* with high (left) and low resolution (right). Bottom row: *Carb-B* with high (left) and low-resolution (right).





Fig 6: Slices of the segmented images used for the numerical simulations to determine permeability and

5   velocities. Top row: *Carb-A* with high (left) and low-resolution (right). Bottom row: *Carb-B* with high (left) and low resolution (right).



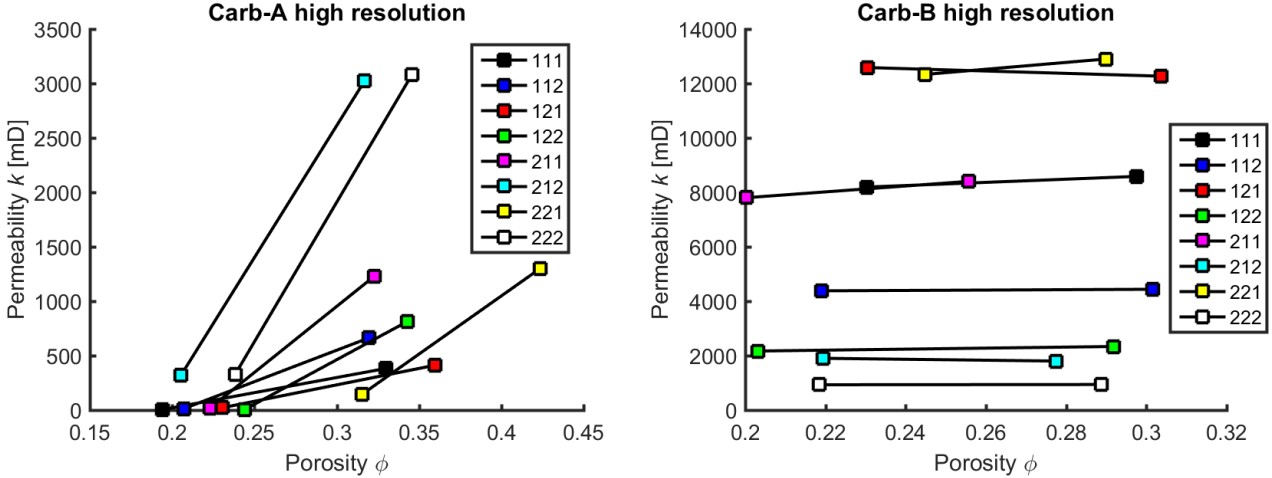

Fig. 7:  Intrinsic permeability for the 8 sub-samples. Results are given for the extreme porosities configuration of the *Carb-A* (left hand side) and *Carb-B* (right hand side) high-resolution samples.




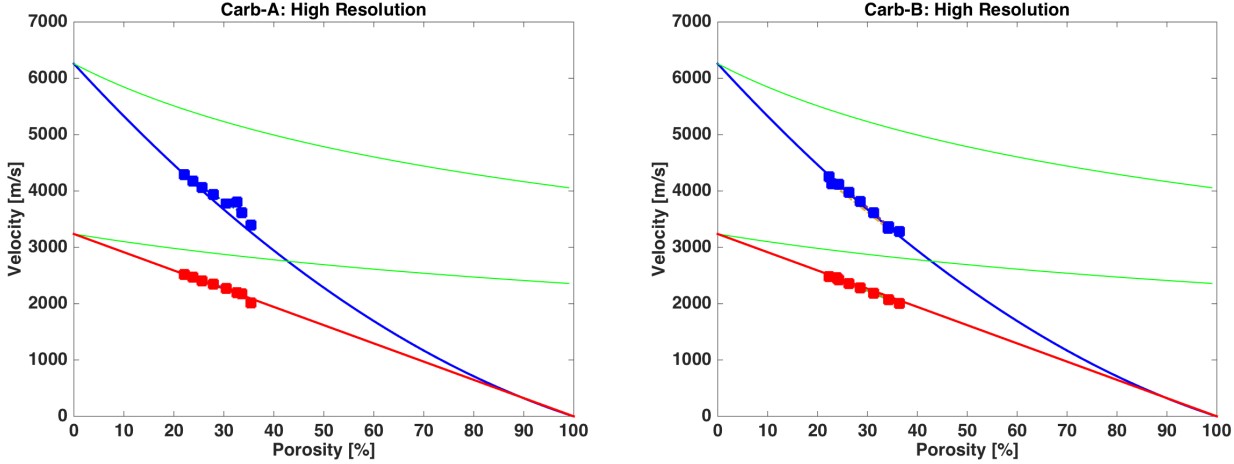

Fig. 8: Effective P- and S-wave velocities (red and blue dots, respectively) for simulations based on a two-phase segmentation. The results were obtained for *Carb-A* (left hand side) and *Carb-B* (right hand side). The blue and red lines are the velocity trends given by equations (4) and (5). The green lines are derived by using the upper Hashin-Shtrikman bounds. For details please refer to the text.





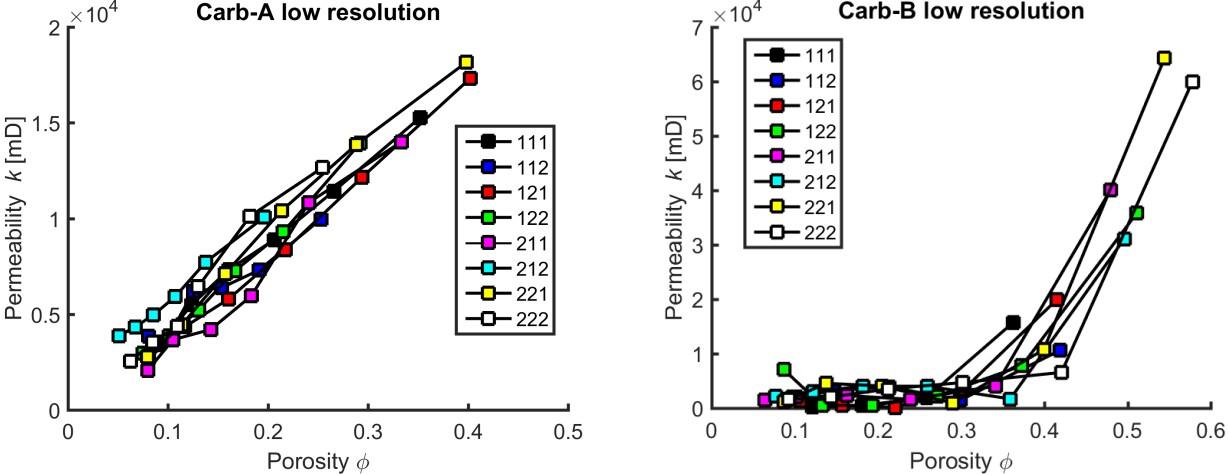

Fig. 9: Intrinsic permeability as a function of porosity. Results for the 8 sub-samples of the *Carb-A* (left hand side) and *Carb-B* (right hand side) low-resolution. Squares markers display each of the pore-solid configurations.





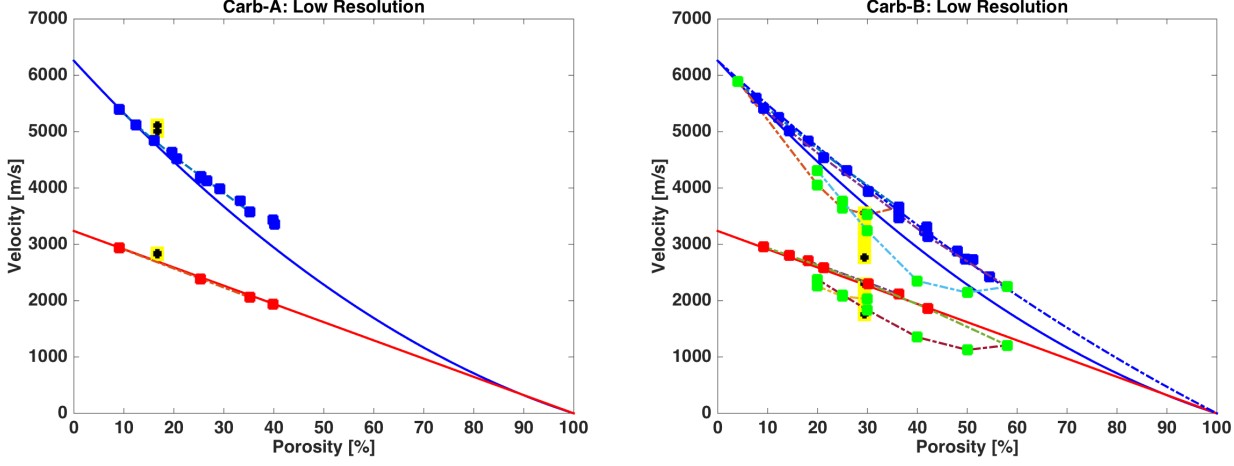

Fig 10: Effective P-wave and S-wave velocities (red and blue dots, respectively) for simulations based on a two-phase segmentation. The results were obtained for the low-resolution images of *Carb-A* (left hand side) and of *Carb-B* (right hand side). The blue and red lines are the velocity trends given by equations (4) and (5); the blue dashed-dotted line is the trend given by equation (6). The experimental results (interval from 0 to 30 MPa confining pressure) are illustrated with black crosses connected with yellow bars for comparison. The green dots display the results of the multi-phase simulations (only performed for *Carb-B* and for p-waves). For details please refer to the text.



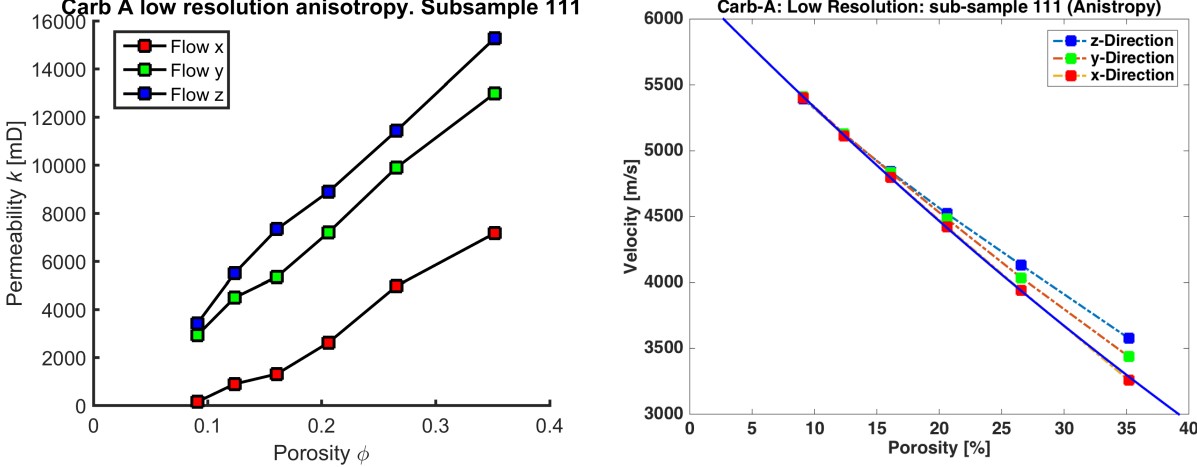

Fig. 11: Left hand side: Intrinsic permeability for sub-sample 111 of *Carb-A* low resolution.
5  Permeability calculated from flow simulated in X, Y and Z directions. Right hand side: P-wave
velocities for all propagation directions of the same sup-sample. The solid line is the velocity trend
given by equation (4). The significant anisotropy visible in permeability is not present for the velocities.



## Tables

Table 1: Helium bulk and grain density (in g.cm$^{-3}$), helium porosity (in PU) and air-permeability (in mD) measured at benchtop conditions, for *Carb-A* and *Carb-B*. Permeability of plug *Carb-A* is below the sensitivity level of the apparatus used (0.1 mD). Values are from Vialle et al. (2013).

| Sample ID | Porosity | Bulk density | Grain density | Permeability |
|---|---|---|---|---|
| *Carb-A* | $0.167 \pm 0.002$ | $2.24 \pm 0.01$ | $2.69 \pm 0.01$ | $< 0.1$ |
| *Carb-B* | $0.294 \pm 0.003$ | $1.90 \pm 0.01$ | $2.70 \pm 0.01$ | $60 \pm 5$ |



Table 2: Pressure dependence of the elastic-wave velocities for the two studied samples. Pressure is in megapascal, and P- and S-wave velocity are in kilometers per second. Values are from Vialle et al. (2013).

|  | Carb-A | | Carb-B | |
| Pressure | Vp | Vs | Vp | Vs |
| --- | --- | --- | --- | --- |
| 0 | 5.007 | 2.835 | 2.769 | 1.754 |
| 2.5 | 5.045 | 2.838 | 3.295 | 2.032 |
| 7.5 | 5.091 | 2.835 | 3.477 | 2.191 |
| 10 | 5.100 | 2.834 | 3.504 | 2.226 |
| 20 | 5.116 | 2.822 | 3.551 | 2.295 |
| 30 | 5.113 | 2.821 | 3.553 | 2.290 |
| 25 | 5.114 | 2.821 | 3.548 | 2.284 |
| 15 | 5.085 | 2.822 | 3.527 | 2.268 |
| 5 | 5.048 | 2.840 | 3.384 | 2.195 |
| 0 | 5.033 | 2.842 | 2.792 | 1.785 |



Table 3: CT-scanner parameters of the acceleration voltage AV and current, respectively. Sample abbreviations are explained in Table 1.

| Pressure | Carb-A | | Carb-B | |
|---|---|---|---|---|
| | Low resolution | High resolution | Low resolution | High resolution |
| Pixel size | 3.4348 μm | 0.6245 μm | 3.4352 μm | 1.1450 μm |
| Image size | 1012 * 1012 | 1012 * 1012 | 1013 * 1013 | 1013 * 1013 |
| Acceleration voltage | 60 kV | 60 kV | 60 kV | 60 kV |
| Current | 80 μA | 80 μA | 83 μA | 83 μA |

