# Peer review of "Digital Carbonate Rock Physics"

_Solid Earth, 2016_

## Referee Comment (RC1) · O. Lopez (Referee) · 31 Mar 2016

The overall quality of the paper is good but a number of minor revisions are needed to make it clearer and ensure good understanding. This paper presents a novel approach for estimating elastic properties of carbonate rocks combining laboratory measurements, imaging technique and simulations. The technique in itself is not new but the authors approach differs regarding the image segmentation to estimate unresolved porosity or unidentified phases and then the measured porosity is used for estimating rock properties. As the editor mentioned, I strongly recommend adding some relevant references regarding previous major works regarding DRP and rock physics: Arns et al., (2002, Geophysics Vol.27), Derzhi et al., (2010, SPE 138586), Ringstad et al., (2013, EAGE) etc... The authors should compare their results with previous published works as per today DRP results regarding elastic properties are rarely matching experimental data and are often overestimating Vp and Vs. Part of the author's technique is

based on image segmentation into different classes. In Page 6, last paragraph should be improved for better understanding. What defines "global thresholds"? Why do you end up with 5 intermediate classes? Please make it more specific. In page 8, first and second paragraph (line 10) should be clearer. It is difficult here to understand the Figure 7 description you made here. How do you end up with a minimum and maximum porosity for each subsample? You mentioned that you selected the "most relevant subsamples" on which criteria is based your choice (porosity only)? A table for both high and low resolution summarizing both calculated and experimental poro/perm will be maybe useful as it is difficult to understand why Figure 7 and 9 are so different. In page 10, Line 18 you highlight the scale issue which is well known from previous studies when comparing DRP and experimental study. Have you made any attempts to upscale DRP results to plug scale? Maybe citation of existing study could be necessary here to avoid misleading conclusions. Paragraph 5.1 you state that even with the highest resolution achievable you cannot resolve all the smallest pores which is true. But techniques exist to overrule these limitations as dry and wet imaging as described by Bhattad et al. (2014, SCA-2014-24) for example. This should be mentioned. Paragraph 5.2 page 12, you write that "porosity values of carbonate using micro-XRCT will only provide estimates with relatively high uncertainty due to significant amount of unresolved pore feature in images". I do disagree with this statement; your approach based on single scale imaging is not suitable for proper porosity estimation. Numbers of published papers show the opposite (Lopez et al, 2012). You should be more specific here and mentioned that for carbonate having one image at a single resolution is not enough for porosity estimation. And this is what you work is supporting, that with a single image and doing some assumptions due to unresolved structures it is still possible to estimate some of the effective properties! Paragraph 5.4, for the Vp and Vs it would have been nice to have the value at infinite resolution as described by Arns et al. (2002) in their Figure 4c. In summary, this paper demonstrates a new way of estimating elastic properties of carbonates containing micritic phases based on micro-XCRT and experimental nano-indentation. This is an elegant way to define moduli that are often

not well known for non-pure minerals and use them for elastic properties determination.

---

## Referee Comment (RC2) · Anonymous Referee #2 · 30 May 2016

The approach of using X-ray micro-CT in carbonates to obtain P- and V-waves is relative new and the idea of selecting distinct threshold values for the frontier between pores and solid matrix is interesting and relevant for the 3D image analysis, presenting a potential addition to the literature dealing with carbonate and digital rock physics (DRP). However, the paper is somewhat written in a confuse way and should be clarified and sharpened throughout the text to improve the understandability of the results. The authors should concentrate more in explaining and discussing their results, instead a big part of the paper deals with the results of other authors with named Tables/Figures (See e.g.: Page 3, line 15; Page 7, line 4; Page 13, line 5) which is inappropriate since results from the literature that the authors refer to should be included in the text of somehow in the paper' structure. Even though it is important, there was a lack of connection between the literature and the authors' own results: what was new from the author's paper compared with the previous literature? The discussion and conclusions were not very clear; for both sections there is a need of pointing out and correlating

the values and results from the tables and figures in the text. In addition, the theory of P- and S-wave velocities applied to the carbonate samples characterization should be elucidated. I suggest a previous definition, its importance related to DRP and the analyzed samples; provide a more detailed discussion between the experimental results and its practical applications. Several comments have been incorporated along the text (see below). Please take them into account (but not limited to) as much as possible.

The title is too general; The Authors should rename the work to better show the focus of their studies.

Abstract: The abstract could be less general also some results (values) should be listed. Page 1, line 15: Please list numbers to the resolutions; line 17: Mention briefly the properties complemented with nano-indentation; line 20: By "intermediate phases" do you mean "intermediate threshold values for distinct phases"? Lines 21-22: This structure is very confusing. To clarify I suggest the authors giving names to the technique/method used in the laboratory to measure porosity, to the predicted effective properties and to the technique used to acquire the experimental data; line 23: Specify that "some sub-samples" actually refers to the distinct smaller regions of interest (ROIs) selected from the acquired CT-datasets. I would also replace "in our case" to "analyzed rocks".

Text: Page 2, lines 7-9: When performing 3D images analysis a helpful tool to investigate and verify the representative elementary volume (REV) of subsamples is using autocorrelation function. Did the authors investigate REV of their samples somehow? The related literature can help: Haussener, S.; Coray, P.; Lipi'nski, W.; Wyss, P.; Steinfeld, A. Tomography-based heat and mass transfer characterization of reticulate porous ceramics for high-temperature processing. ASME J. Heat Transf. 2010, 132, 023305:1–023305:9. Petrasch, J.; Wyss, P.; Stämpfli, R.; Steinfeld, A. Tomography-based multiscale analyses of the 3D geometrical morphology of reticulated porous ceramics. J. Am. Ceram. Soc. 2008, 91, 2659–2665. Haussener , S.; Steinfeld, A. Effective Heat and Mass Transport Properties of Anisotropic Porous Ceria for Solar Thermochemical Fuel Generation. Materials 2012, 5, 192–209. Costanza-Robinson, M.S.; Estabrook, B.D.; Fouhey, D.F. Resentative elementary volume estimation for porosity, moisture saturation, and air-water interfacial areas in unsaturated porous media: Data quality implications. Water Resources Research, 47, WO7513:1–WO7513:12. Bear, J. Dynamics of Fluids in Porous Media, General Publishing Company LTD, 1972. pp. 19–21.

Page 2, lines 11-13: In which type of material/rock? This statement can be invalid e.g., when analyzing other rock types such as shale; line 16: "3D rock models"? Maybe, "3D rock pore networks". Line 19: It is not the porosity which is smaller, but the pore sizes; lines 20-22: Once more authors draw a statement which is in fact strongly depending on the material/rock type and acquired voxel resolution. Please add rock type and resolution range to correct sentence; line 28: rephrase sentence.

Page 3, line 2: Take out "as well"; lines 3-6: How did the authors managed to improve "digital rock images themselves and/or the computational workflow"? Describe it succinctly relating e.g., image enhancement with image acquisition parameters, voxel resolutions, pre- and post-processing; line 4: Correct the verb form; line 6: name the "suggested techniques"; line 7: Complementary in which aspects? Authors should use this structure to point out in more details the importance of their work and in which aspects it is novel and relevant compared with the former cited studies.

Pages 3-4, lines 26-7 and Tabs. 1 and 2: Remember using S.I. standard units and note that the numerical value precedes unit and a space is always used (except for degree, minute, and second for plane angle) to separate them.

Subtitles are too short and should be improved making a least description of each subsection.

Page 4, line 14: "(RMS values)" should be moved to right after "1.4 $\mu$m".

Page 5, lines 12-16: Authors mentioned Poisson's ratio but do not say which values

they used for their calculation? Fig. 1 is under explained, e.g., the blue and green areas mentioned in the caption should be clarified in the text; Explain also which is the relation/implication between the "blue and green" areas and the nano-indentation results. Clarify the real mean/relevance of Fig. 1 to the paper context as well.

Page 6, line 1: Inform the source-to-sample distances in Table 3 and change "pixel size" to "voxel size" adding the cubic unit to the values as well; line9: "illuminate"? Line 12: I suggest the authors take out Fig. 2 and only present this sequence in the text itself, since this workflow is relatively simple and brings no novel information to the paper; line 13: Give the voxel size of selected ROI. Fig. 4: Authors should describe the dark green areas, which are overlapping volumes between neighbors subsample ROIs, to improve understanding of their procedure; line 18: Keep a standard on typing: "subvolumes" or "sub-volumes", "subsamples" or "sub-samples"; Lines 20-22: "appropriate dimensions and kernel window sizes" which were?

Page 7, line 7: Were the same Carb-A and Carb-B samples investigated by Vialle et al., 2013? In positive case, I suggest the authors to add the values of Hg porosity and compare it succinctly to the He porosity (shown in Table 1) and distinct CT porosities obtained from the thresholds levels of micritic phases. This will give an idea of the optimal threshold value which is surely related to the effective rock properties moreover discussed in the work. Line 14-15: give the used values for pressure bound condition and dynamic viscosity of fluid; line 19: what does the form "RSG" stand for? Note that Fig. 6 was not commented in the manuscript text. If Fig. 6 isn't that relevant to the paper' findings it should otherwise be taken out.

Page 8, line 6: What do you mean by "most relevant subsamples"? Give the criteria to judge a subsample relevant; also do the authors mean by "numerical investigation" in this structure the P-and S-waves velocities? Please clarify! Because if one looks to the numerical investigation of permeability (Figs. 7 and 9) it is possible to see that simulations were performed in all 8 subsamples, while P-and S-waves velocity simulations are given only for one subsample (give the subsample names in the legend) of each

carbonate (Figs. 8 and 10). In Fig 7, add "simulated" after "Intrinsic permeability" and in the graphic axis (in Fig. 9 as well). Comparing the results of permeability simulations for the high resolution (Fig. 7) and the low ones (Fig. 9) one can see that only the minimum and maximum threshold values were depicted in Fig. 7. Please elucidate the reasons for that. Lines 11-18: Authors made a good observation and should justify this result better. Another interesting find when comparing Figs. 7 and 9 is the variation on the permeability results between subsamples: for the low resolution results less variation in the permeability is observed compared to the higher resolution, indicating less anisotropy of the subsamples and more material representativity. It is an important find in your study, you have it in numbers and you should highlight it! Observe as well how the subsamples of Carb-B (high resolution) showed to be heterogeneous; even though as the authors describe "it shows a much lower variation between the extreme values", the subsamples have extreme variation in the permeability values compared with subsamples of Carb-A. Which would be the probable causes for these results? Line 18: This statement is half wrong! Line 20: Here the "micritic phases" term is given without a clear explanation that they actually are the distinct phases identified from the threshold' classes of 3D images (as described in section 3.2). Please clarify it also linking it to the Fig. 5. The same is happening in section 4.2.1 when a new term "six possible domains" is introduced.

Page 9, line 12: In fact the threshold values are being varied what implies in the porosity change! Lines 18-19: Make sure to inform that these results are shown in Fig. 11; lines 22-23: Rephrase structure; lines 25-26: Rephrase the position of "(Figure 10)" in the structure.

Page 10, lines 1-4: I disagree that only Carb-B showed slightly difference, which can also be seen in the P-wave results of Carb-A, on which data "a blue dashed-dotted line" should be fitted as well. IMPORTANT: Note that if P-waves are represented with the blue color in Figs 8 and 10, captions must be corrected. Lines 13-16: The performed procedure and described results are very interesting for a better discussion; lines 21-

23: Make a link to it commenting the finds from Barb-B (Fig.7) and discussing in a practical manner how the present work overcomes this problem.

Page 11: Section 5.1: Although the idea of correlating estimated elastic properties of carbonates based on distinct micritic phases identified from the threshold' levels in micro-XCRT images, with experimental nano-indentation experiments sounds very attractive, the authors failed in their expectations described in the lines 22-24. For a rock/material having a defined amount of pores and solid matrix, one can expect an image threshold with at least two peaks: one in the darker gray levels regions (related to the pores) and another in the brighter regions (linked to the matrix); however If the analyzed material has also a certain amount of heavy phases (e.g. iron) then another additional peak in the threshold can be observed. Whereas (as the authors described very well) it is difficult to see the moduli peaks of pores in the nano-indentation experiment results, naturally because the values are very low. The relation from the micro-XRCT images and nano-indentation experiments using the number of threshold peaks seems somehow inappropriate.

Page 12, lines 5-8: Include figures numbers (low and/or high resolutions) of your work to improve reading and understanding; line 16: nane the technique used to the measured porosity or add "as shown in Table 1". In line 18: specify "full sample". Lines 22-24: Authors should be careful and add in this statement, that this observation is for their specific case (Carb-A and Carb-B) within the investigated resolutions which is based on the single image scales. Nowadays the use of multi-scale approaches to investigate porosity and DRP of heterogeneous rocks such as carbonates became widely common and has proving to be reliable.

Page 13, lines 2-3: name the tables/and figures from were readers can see these results; line 9: change to "experimental measurement". Line 12: name the porous materials; lines 15-16: How "statistically significant" (also given the Summary) samples should be? Try to base it on your results with the proposed approach using multi micritic phases and subsamples (ROIs).

Page 14, section 5.5: the statement that "any significant anisotropy for permeability" was found in the analyzed samples is in disagreement with some of the paper' results (see e.g. Fig. 7). Elucidate the anisotropy changing from the higher to the lower resolutions, more evident for Carb-B than Carb-A.

Concerning to the Summary: Summary is in general written in a confuse way making it hard to follow the author's thoughts. The summary should be rewritten in a more focused and brief way. Again, the authors provide their conclusions without backing them up with the quantified values that they base their assumptions on, making the work appear somewhat subjective. They tend to loose themselves in generalizations such as "the porosity of the rock samples is the most relevant parameter"; certainly the authors do not mean that for any purpose in the world including rocks porosity is the most relevant parameter, as an example for structures that need to be sharpened and detailed.

Several references are missing, i.a.: Page 2, lines: 10-11, 14-15, 15-18, 22-24; Page 4, line 22; Page 5, lines 3, 7; Page 6, line 8 (reference the model used in the reconstruction); Page 6, line 20; Page 7, lines 7, 10; Page 13, line 27.

―――――――――――――――――

---

## Author Comment (AC1) · 7 Jul 2016

Please find our answers to the comments and the revised manuscript in the Supplement.

Please also note the supplement to this comment:
http://www.solid-earth-discuss.net/se-2016-45/se-2016-45-AC1-supplement.pdf
* * *

---

## Author Comment (AC2) · 7 Jul 2016

**Digital Carbonate Rock Physics**

Erik H. Saenger[1,2], Stephanie Vialle[3], Maxim Lebedev[3], David Uribe[2,4], Maria Osorno[2,4], Mandy Duda[1], and Holger Steeb[4,5].

[1]International Geothermal Centre, 44801 Bochum, Germany
[2]Ruhr-Universität Bochum, 44801 Bochum, Germany
[3]Curtin University, Perth, Australia
[4]University of Stuttgart, 70569 Stuttgart, Germany
[5]Stuttgart Research Centre for Simulation Technology (SRC SimTech), 70569 Stuttgart, Germany

*Correspondence to*: E. H. Saenger (erik.saenger@rub.de)

Keywords: Carbonates; Effective elastic properties; Permeability; Digital Rock Physics; Nano-indentation;

**Abstract.** Modern estimation of rock properties combines imaging with advanced numerical simulations, an approach known as Digital Rock Physics (DRP). In this paper we suggest a specific segmentation procedure of X-Ray micro-Computed Tomography data with two different resolutions in the μm-range for two sets of carbonate rock samples. These carbonates were already characterized in detail in a previous laboratory study which we complement with nano-indentation experiments (for local elastic properties). In a first step a non-local mean filter is applied to the raw image data. We then apply different thresholds to identify pores and solid phases. Because of a non-neglectable amount of unresolved micro-porosity ("micritic phase") we also define intermediate threshold values for distinct phases. Based on this segmentation we determine porosity-dependent values for effective P- and S-wave velocities as well as for the intrinsic permeability. For effective velocities we confirm an observed two-phase trend reported in another study using a different carbonate dataset. As an upscaling approach we use this two-phase trend as an effective medium approach to estimate the porosity-dependent elastic properties of the micritic phase for the low-resolution images. The porosity measured in the laboratory is then used to predict the effective rock properties from the observed trends for a comparison with experimental data. The two-phase trend can be regarded as an upper bound for elastic properties; the

use of the two-phase trend for low-resolution images lead to a good estimate for a lower bound of effective elastic properties. Anisotropy is observed for some of the considered subvolumes, but seems to be insignificant for the analysed rocks at the DRP scale. Because of the complexity of carbonates we suggest to use DRP as a complementary tool for rock characterization in addition to classical experimental methods.

**1 Introduction**

Three-dimensional information on rock microstructures is important for a better understanding of physical phenomena and for rock characterization on the micro-scale. Various destructive and non-destructive methods for obtaining a 3D image of the rock microstructure exist (Arns et al. 2002; Saenger et al. 2004; Madonna et al., 2013, Cnudde and Boone, 2013, and references therein). The most common non-destructive 3D imaging technique for rock samples is X-Ray Computed Tomography (XRCT). A common problem, however, is a clear trade-off between sample size and resolution. For each material a specific, and large enough, sample size is required to ensure that the selected volume is representative of the physical property to be computed (e.g., Hill 1963; Costanza-Robinson et al. 2011; Andrä et al. 2013a). It can, however, be at the expense of a lost of pore features resolution. In the last decade, the X-Ray micro-Computed Tomography (micro-XRCT) method became widely available and many modern studies have made use of it to obtain 3D rock images (e.g., Fusseis et al. 2014). The resolution of micro-XRCT of up to $(0.6 \ \mu m)^3$ (voxel size) is high enough to image the spatial distribution of grains, pores, and pore fluids for a wide variety of rocks (e.g., Blunt et al. 2013; Madonna et al. 2013).

Rock images from 3D XRCT can be used for predicting properties such as porosity, permeability, pore size distribution, effective elastic moduli, or electrical conductivity (e.g., Andrä et al. 2013b). For example, Darcy permeability can be predicted by numerically simulating single-phase fluid flow through 3D rock pore structure models, with the numerical results being in reasonable agreement with laboratory measurements (e.g., Osorno et al. 2015, Saenger et al. 2016). In this case, the resolution of the micro-XRCT technique is sufficient, because fluid pathways predominantly follow larger pores. However, if the pore size is much smaller than 1μm the agreement might be less satisfactory due to

resolution limitations. On the other hand, mechanical properties, such as the effective elastic moduli, strongly depend on the microstructural details of the rock, which may stay unresolved by the micro-XRCT technique (e.g. for Bentheim sandstone considered in Saenger et al., 2016). The inability to fully characterize the microstructural details of a rock sample can lead to a disagreement between numerical estimates of mechanical properties based on micro-XRCT images and laboratory data (Andrä et al. 2013b).

An example of such a disagreement between laboratory and Digital Rock Physics (DRP) estimates is described in Andrä et al. (2013a,b). In these benchmark papers a comparison between different numerical methods is presented. All DRP estimates of the effective elastic bulk modulus use the same segmented dataset. Regardless of the numerical approach all computational predictions overestimate the bulk modulus measured in the laboratory. This conclusion is mostly based on Berea sandstone although carbonates are considered in this study. However, also Jouini et al. (2015) reports about an overestimation of effective elastic properties of carbonates by DRP. Therefore we conclude here that the digital rock images themselves and/or the computational workflow have to be improved to provide better estimates of effective properties of rocks. In this paper we consider in detail a carbonate dataset and suggest techniques to achieve a better agreement between numerical predictions and laboratory measurements. Our study is complementary to the DRP-carbonate studies performed in Derzhi et al (2010), Lopez et al. (2012), Ringstad et al. (2013), Andrä et al. (2013a,b) and Jouini et al. (2015). In contrast to these studies our Digital Rock Physics study is complemented with a very detailed experimental characterization (section 2). Our suggested segmentation technique (section 3) is used to estimate effective mechanical as well as effective transport properties (section 4). Among others, we observe a two-phase trend for mono-mineralic (calcite) carbonates which can be regarded as an upper bound for velocities at all scales (see discussion in section 5) due to the observed self-similarity of those rocks (Jouini et al. 2015).

**2 Rock Samples and Laboratory Characterization**

**2.1 Carbonate samples**

[revised manuscript text omitted]

Young's moduli , *E*, can be calculated from the indentation moduli (Fischer-Cripps 2004) according to

$$\frac{1}{M} = \frac{1-v^2}{E} + \frac{1-v_i^2}{E_i} \quad . \quad (2)$$

Indenter properties are $E_i$= 1220 GPa and $\nu_i = 0.06$, according to Klein & Cardinale (1992) and Fischer-Cripps (2004), for diamond material. Each performed measurement covers a projected surface of about 40µm² on average (which corresponds to an equilateral triangle with a side of 6µm), and contains both pores and solid grains. A Poisson's ratio has also to be assumed for each nano-indentation measurement: even though we can compute it from the laboratory ultrasonic P- and S- wave measurements at the core scale, there is no reason to assume that this value is constant for each individual measurement. This local Poisson's ratio cannot be measured experimentally and we have taken here a constant value of 0.3. Figure 1 displays the distribution of the indentation moduli for both *Carb-A* (left) and *Carb-B* (right).

**3 The CT-datasets**

**3.1 Procedure to get CT-Images**

Two samples were prepared for imaging with micro-XRCT from the cuttings, one from *Carb-A* and one from *Carb-B*. A cylindrical-shape sample of 1.5cm in height and 2mm in diameter was achieved by gently grinding the cuttings, first on the side on a rock saw blade, and then by hand using sand paper (grit 120). This procedure allows obtaining very thin cylinders while minimizing mechanical damage that classical drilling would produce. These cylindrical samples were then glued with Crystalbond509 (SPI suppliers) on a 2mm diameter flat-head metal pin, which was itself inserted in the core holder of the micro-tomograph. The 3D X-ray Microscope Versa XRM 500 (Zeiss - XRadia) was used with a X-ray energy of 60 keV. Two different settings of source-to-sample and detector-to-sample distance were used to achieve two nominal voxel sizes of $(3.4 \ \mu m)^3$ and $(0.6 \ \mu m)^3$ for *Carb-A*, and of $(3.4 \ \mu m)^3$ and $(1.1 \mu m)^3$ for *Carb-B*, referred to as "low" resolution and "high" resolution, respectively, for each for the two samples. X-ray microscope and image acquisition settings are summarized in Table 3 for each of the two samples.

The number of radiographic projections acquired during sample imaging with "low" and "high" resolutions were 3001 and 5001, respectively. The total scanning time for one sample was about 8h. Initial cone-beam 3D image reconstruction was performed using the internal software XM

Reconstructor (XRadia). To remove geometrical artefacts during reconstruction a secondary reference was acquired for samples image with maximum resolution.

**3.2 Segmentation procedure**

In addition to solid grains and pore space different micritic phases are visible in the raw images of the scans entailing an advanced segmentation procedure. For our segmentation (Figure 2) we select a region of interest (ROI) from the raw data of the two types of carbonates with two different resolutions (Tab 3). The ROI is subdivided in eight partly overlapping subvolumes, each of a size of $400^3$ voxels (Figures 3 and 4). For the low-resolution images, it gives thus eight cubes with a side of 1.37 mm for both samples, and for the low-resolution images, 8 cubes with a side of 0.25 mm for *Carb-A* and of 0.46 mm for *Carb-B*.

Our segmentation workflow is applied to the full ROI including all subvolumes. Image enhancement and segmentation steps were carried out using the software package Avizo Fire 9 (FEI Visualization Sciences Group). Before actual segmentation the image noise and scan artefacts are reduced while preserving interfaces using a 3D non-local mean image filter. To our experience the standard values of this filter are appropriate (Search Windows = 21;  Local Neighborhood = 5; Similarity Value = 0.6). Note that every step of image enhancement changes the original data set affecting subsequent steps required for data analysis.

The image-enhanced datasets were segmented into classes using global thresholds for the covered range of grey-values. The global threshold is valid for all the eight partly overlapping subvolumes mentioned above. Considering that the samples represent quasi monomineralic calcitic rocks we identified the following classes illustrated in Figures 5 and 6:

- high-confidence pores (illustrated with dark blue color),
- high-confidence mineral (illustrated with dark red color),
- and five intermediate classes.

Note that a non-negligible part of the pore space is below the resolution limit of the µ-CT scans (cf. results of mercury intrusion showing pores and pore throats as low as 0.06 µm, Figure 6 of Vialle et al. (2013). We found that five intermediate classes are sufficient to describe the calcitic rocks used in this

study, but the number of classes between the high-confidence mineral class and the high-confidence pore class can be increased or decreased for other materials. In contrast to Lopez et al. (2012) or Ringstad et al. (2013) we do not think that the density can be approximated directly by the grey value of those micritic phases. From our point of view the relationship between grey-value and density can be highly non-linear and becomes even more complicated in the case of multi-mineral rocks. However, we make use of the accepted assumption that lighter grey values in the histogram correspond to lower porosities.

**4 Numerical Results**

In order to numerically calculate the effective intrinsic permeability $k$ of the digitized rock sample we calculate the fluxes under creeping flow condition based on a parallelized Stokes-solver. The parallelized Finite Difference-based Stokes-solver is suitable for the calculation of effective hydraulic parameters for low and high porous materials (cf. Osorno et al., 2015). Using volume averaging technique, we coarse-grain the local velocity field $u(x)$ obtaining the global velocity component $u_m$ in flow direction.

The intrinsic permeability $k$ is calculated with Darcy's law:

$$k = \frac{\mu\, u_m}{\Delta p}. \qquad (3)$$

The pressure gradient $\Delta p$ is imposed with pressure boundary conditions in the numerical simulations. The dynamic viscosity of the pore fluid is $\mu$. In our numerical simulations $\Delta p$ is -5.8 10-4 Pa/m and $\mu$ is 1.2 Pa.s.

To obtain effective P- and S-wave velocities of the digitized rock samples we use a technique described in detail in Saenger et al. (2011) and references therein. The basic idea of this approach is to study speeds of elastic waves through heterogeneous materials in the long wavelength limit (pore size $\ll$ wavelength) using the rotated staggered grid (RSG) finite-difference algorithm (Saenger et al., 2000) for the simulation of elastic wave velocities (cf. Andrä et al. 2013a,b).

**4.1 High Resolution**

**4.1.1 Permeability**

Permeability calculations were realized for subvolumes of the *Carb-A* and *Carb-B* samples. However, the domain size of the *Carb-A* (0.43 mm) and *Carb-B* (0.78 mm) high resolution samples is smaller than

5  the low resolution ones (2.4 mm for *Carb-A* and *Carb-B*), i.e. less representative of the material, therefore we numerically investigate only the extreme porosity configurations. To select the domains for permeability calculation we adapt the porosity configurations (range defined by high-confidence pores to high-confidence grains; see also discussion in section 5.2) showing the minimum and maximum deviation in porosity with respect to the experimental investigation. To analyse the

10  homogeneity of the sample this step was performed for all 8 subvolumes. For the high resolution subdomains (*Carb-A* and *Carb-B*) we perform Stokes flow simulations only in one direction (z-direction, cf. Figure 3).

Figure 7 left hand side displays the intrinsic permeability calculated for the *Carb-A* high resolution. The porosity range of the subvolumes is higher than the experimentally determined porosity. In addition, the

15  numerically calculated permeability values are significantly lower than the values obtained for the low-resolution samples. Figure 7 right hand side shows the results of the intrinsic permeability calculated for the selected high-resolution samples of the *Carb-B*. It can be observed, that the high-resolution sample show a much lower variation between the extreme values of the porosity range.

From the results of the high-resolution samples, *Carb-A* and *Carb-B,* it could be observed that the

20  variation in pores channels arrangement is significant and the permeability in the different subvolumes of the same material does not necessarily increase with the porosity increment.

**4.1.2 Elasticity**

Several micritic phases have been identified in the raw images of the carbonate rock (i.e. the phases between high-confidence pores and high-confidence minerals; compare with section 3.2). The porosity

25  of these regions cannot be determined exactly, as some pores are below the resolution of the scans: typically, micrites exhibit pore sizes with a maximum diameter of 1 μm (Moshier, 1989; Cantrell and Hagerty, 1999), and pore sizes as low as 0.06 μm have been measured by MICP for the samples under

investigation (Vialle et al., 2013). To account for the unresolved pore space we perform a number of two-phase wave-propagation simulations to estimate effective elastic properties (Saenger et al. 2004; Saenger et al. 2011; Saenger et al. 2016). For these time-of-flight simulations we assign vacuum properties to the pore phase, while the rest of the digital image including the micritic phase will be

5   assumed to be solid with the mineral properties of calcite (e.g. Andrae et al., 2013b). For the second simulation, we assign vacuum properties to the pores and the first micritic phase, while the rest will be assumed to be solid. We continue this way for all micritic phases, so that the last simulation assigns the mineral properties of calcite only to the high-confidence mineral phase. By this technique we obtain a porosity-velocity trend (Figure 8) for a random selection of high resolution subvolumes for *Carb-A* and

10   *Carb-B*:

$$Vp = 6259.1 \text{ m/s} - \phi * 9640 \text{ m/s} + 3381 \text{ m/s} * \phi^2 \qquad (4)$$

$$Vs = 3237.2 \text{ m/s} - \phi * 3237.2 \text{ m/s} \qquad (5)$$

This porosity-velocity trend is exactly the same as has been observed for a carbonate dataset from a different location used in Saenger et al. (2014). In their paper this trend has been observed for three

15   different resolutions (65 nm, 1 μm, 4 μm). Please note that due to computational restrictions we are only able to simulate a random selection of subsamples; however, as shown in Figure 8, all our calculated velocities follow the trends according to equations (4) and (5).

**4.2 Low Resolution**

**4.2.1 Permeability**

20   Similar to the procedure of the numerical simulation for elasticity (Section 4.1.2) we vary the sample porosity. This way we get six different porosities for each subvolume depending on the threshold variation. To reduce computational times for the Stokes flow simulation we eliminate the disconnected pores. Some subvolumes solid-pore configurations with the lower porosities do not present connected pores, and we assume for the effective permeability *k*=0.

25   Figure 9 present the permeability values for *Carb-A* (left hand side) and *Carb-B* (right hand side) samples as a function of porosity. For the permeability calculations for these samples we perform Stokes flow simulation in z-direction only (compare Figure 3).

Additionally we performed the Stokes flow simulations in three directions (X, Y and Z) for *Carb-A* sample (see Figure 11). From the simulation results it can be seen that the *Carb-A* sample permeability (Figure 11 left hand side) is anisotropic with a variation between directions of up to 80 %. In some subvolumes permeability value varies by up to two orders of magnitude.

5  From the CT data of the low resolution *Carb-A*, the largest domain that could be extracted is 2.4 mm x 2.4 mm x 2.4 mm ($689^3$ voxels). The permeability calculated for this domain is 13.0 D for a porosity of $\phi = 0.173$.

**4.2.2 Elasticity**

For the low-resolution scans we repeat the two-phase simulations for *Carb-A* and *Carb-B* as described
10  in section 4.1.2. The results are displayed in Figure 10. Interestingly, the two-phase trend given by Equations (4) and (5) is confirmed only clearly for S-wave velocities of *Carb-A* and *Carb-B*. For the case of P-wave velocities we observe a slightly different trend (cf. eq. (4)) that we illustrate with blue dashed-dotted lines in Figure 10:

$$Vp = 6259.1 \text{ m/s} - \phi * 7970 \text{ m/s} + 1700 \text{ m/s} * \phi^2 \qquad (6)$$

15  Especially in the low-resolution case we expect to have images with a large amount of unresolved porosity, mainly due to micritic phases. Therefore we perform multi-phase simulations and vary the porosity by assigning effective elastic properties to an interval of micritic phases (always starting with the class closest to the high-confidence pore phase). As effective medium approach we use the trend given by the simulations using two single phases only (Equations (4) and (5)), which is supported by
20  two observations. First, this trend was already observed by Saenger et al. (2014) on different scales on a different carbonate dataset. Second, there is an observed self-similarity for carbonates (Jouini et al. 2015). Therefore, despite the interval of micritic phases, we assign vacuum values for the high-confidence pores and use for the remaining phases the known elastic moduli of calcite (e.g. Andrae et al., 2013b). The results are displayed with green dots for *Carb-B* in Figure 10. We repeat the procedure
25  with different intervals of the micritic phases. There are three interesting observations: (1) the resulting effective velocities are always significantly below the observed two-phase trend, (2) the curves for

different intervals will intersect each other, and (3) the experimental determined velocities for high confining pressures are between the multi-phase results and the two-phase trend as described above.

**5 Discussion**

In this paper we compare results from laboratory investigations with numerical estimates based on digital images. Note that in laboratory experiments we use samples on the cm-scale for the determination of permeability and ultrasonic velocities and compare it with DRP-predictions based on images on the mm-scale. Especially because of the known heterogeneity of carbonates there is always a risk that the selected scanned area is not representative compared to the full sample size used for laboratory characterizations. In general, a multi-scale approach as suggested by Ringstad et al. (2013) should be used for upscaling the results to the plug scale. However, our studies on *Carb-A* and *Carb-B* suggest workflows which should be applied in practice for as many samples as possible for improving the statistical significance.

**5.1 Discussion of Experimental Characterization**

Even with the highest resolution currently available in micro-XRCT imaging there will be a significant amount of unresolved pore features which need to be treated in the DRP workflow (Saenger et al., 2016). On the grey-scale intensity level histograms of the low- and high-resolution images of the micro-CT scanning (Figure 5) this is reflected in a continuum in the intensity levels between the phase identified as pores and the phase identified as calcite grains. In this paper we have dealt with these micritic phases by replacing, step-by-step, and in a cumulative way, each of the micritic phases by void, and establishing a porosity-velocity trend. A more advanced technique using dry and wet imaging is suggested by Bhattad et al. (2014)using the difference imaging to approximate effective properties. However, the nano-indentation technique provides a measure for the distribution of effective elastic properties at the micrometer-scale, and can thus potentially constrain the input parameters for the different phases identified during the segmentation. To be able to do so, nano-indentation needs to provide bulk and shear moduli from each of the measurements (load-displacement curves) and we need to obtain effective bulk and shear moduli values for each of the identified phases in the microtomography (pores, calcite grains and the five micritic phases). However, if nano-indentation technique is a well-established technique in material sciences, which deals with homogeneous, purely elastic materials, this is, as of today, not yet the case for rocks, which are heterogeneous materials with both elastic and non-elastic behaviour (creep). Though nano-indentation tests provide significant insights

5 into elastic properties of heterogeneous rocks such as carbonates (Lebedev et al., 2014; Vialle & Lebedev, 2015) or shale (Ulm & Abousleiman, 2006; Abousleiman et al., 2007), there are still some points to be looked at before using the derived values of Young's (or shear and bulk) moduli in a quantitative way for DRP: value of Poisson's ratio to be used, effect of surface roughness, local mechanical damage induced on the sample's surface by polishing techniques, etc. Nonetheless, the

10 histograms of the indentation moduli of both samples show a broad distribution of moduli values ranging from very low values (a few GPa, where the indenter tip measures stiffness of an area mostly made of a pore) to values consistent with calcite. The existence of these intermediate values is consistent with the existence of micritic phases identified with X-ray tomography. However, we did not observe two peaks in the histogram for the nano-indentation results (Figure 1) in contrast to the

15 histograms of the scanned micro-XRCT-images (Figure 5). Therefore the direct translation of moduli derived from nano-indentation remains also difficult. The "resolution" of nano-indentation used in this study allows for determining effective elastic properties at slightly bigger scales than that used here for the micro-XRCT.

Regnet et al. (2014) showed that there is a relationship between micrite microstructure and laboratory

20 ultrasonic velocities on core samples, with samples with higher content of 'tight micrite' exhibiting higher velocities and samples with higher content of 'microporous micrite' exhibiting lower velocities. Studied core samples were though a mixture of different types of micrite and the measured velocities represent effective properties at the core scale. This observation is reflected in the established porosity-velocity trends (Equations 4, 5 and 6): micritic phases with density closer to that of calcite ('tight

25 micrites') have higher velocities than micritic phases with lower density (Figures 8 and 10) closer to that of pores ("microporous micrites').

**5.2 Discussion of Porosity: Experiments vs. Digital Rock Physics**

After the segmentation it is also possible to estimate the porosities of the samples. Based on the suggested workflow described in section 3.2 there will be a lower and upper bound. For the lower bound we will treat only the "high-confidence pores" as pores; for the upper bound we count only "high-confidence minerals" as minerals.

[revised manuscript text omitted]

**6 Summary**

With the current imaging techniques it remains difficult to resolve microstructures (on sub-micrometer scale) and image a representative volume at the same time, which is essential to understand the effective material properties of rocks. For this purpose the exact determination of the porosity of the rock samples is the most relevant parameter. To overcome this problem, we have conducted a specific multi-phase segmentation technique and a careful calibration of DRP estimates with laboratory data. Especially for carbonate samples it is difficult to estimate exactly the porosity from raw-CT data, because the micritic phases remain unresolved with an unknown porosity. Therefore, we use our presented numerical results in an inverse way. We suggest using the porosity determination from the laboratory and go back to our low-resolution trends given in Figure 9 and 10. With a given porosity we can now estimate the permeability and the effective wave velocities.

In case of the studied samples *Carb-A* and *Carb-B*, we can predict P- and S-wave velocities with a good agreement to laboratory results. The presented two-phase trend (Equations 4, 5 and 6) is found to be an upper bound for a wide range of scales and can also be used as an effective medium approach to the micritic phases. The predicted permeability values are only in good agreement for *Carb-B*. Most probably the low-resolution image of *Carb-A* is not representative for the sample used in the laboratory. However, for the used carbonate rock samples anisotropy seems insignificant for elastic as well as for hydraulic properties.

In general, the resolution of the XRCT is the limiting factor for the application of DRP for carbonate rock. The micritic phases remain unresolved even for the highest resolutions available. Therefore, the effective elastic properties have to be approximated. Our suggestion is to use the trend of the two-phase

simulations. The implemented workflow in this paper can be applied in general for numerical estimates of mechanical and transport properties of carbonates. Because of the known strong heterogeneity of carbonates we suggest to use a statistically significant amount of digital images to characterize a formation.

**5 Acknowledgements**

Erik H. Saenger would like to thank ExxonMobil for the support of some ideas presented in this study. This work was partially funded by Curtin Reservoir Geophysical Consortium (CRGC). The authors thank the National Geosequestration Laboratory (NGL) of Australia for providing access to the X-ray microscope Versa XRM 500 (Zeiss - XRadia) and to the Nanoindentation system (Fisher-Crips Laboratories Pty.Ltd.). Funding for this facility was provided by the Australian Federal Government.

**Figures**

[Figure]

5    Fig. 1: Nano-Indentation results for *Carb-A* (left hand side) and *Carb-B* (right hand side). In blue we illustrate the corresponding moduli range from ultrasonic experiments on dry samples from 0 to 30 MPa confining pressure, and in green we illustrate the moduli range given by the solid anisotropic calcite crystal. Overall we observe that the medium effective indentation module M is slightly stiffer for *Carb-A* (26 GPa vs. 25 GPa).

[Figure]

Fig. 2: Simplified segmentation workflow as applied in this study.

[Figure]

Fig. 3: Sketch to illustrate the segmentation geometry (here for *Carb-B*; high-resolution). The full cube is subdivided into eight partly intersecting subvolumes with a size of $400^3$ gridpoints. Those subvolumes are used in the numerical simulations to estimate effective material properties.

[Figure]

Fig. 4: Slices of the raw tiff-images of the scanned samples. The dark green areas mark the overlapping zones of the considered subvolumes. Top row: *Carb-A* with high (left) and low resolution (right). Bottom row: *Carb-B* with high (left) and low resolution (right).

[Figure]

5   Fig 5.: Color-coded histograms of the scanned samples. Top row: *Carb-A* with high (left) and low resolution (right). Bottom row: *Carb-B* with high (left) and low-resolution (right). Between the high-confidence pore phase (marked blue) and the high-confidence mineral phase (marked red) we define five intermediate classes to characterize the micritic phases within carbonate rock.

[Figure]

Fig 6: Slices of the segmented images used for the numerical simulations to determine permeability and velocities. Top row: *Carb-A* with high (left) and low-resolution (right). Bottom row: *Carb-B* with high (left) and low resolution (right).

[Figure]

Fig. 7: Intrinsic permeability simulated for the 8 subvolumes. Results are given for the extreme porosities configuration of *Carb-A* (left hand side) and *Carb-B* (right hand side) high resolution samples.

[Figure]

Fig. 8: Effective P- and S-wave velocities (red and blue dots, respectively) for simulations based on a two-phase segmentation. The results were obtained for a random selection of subvolumes of *Carb-A* (left hand side) and *Carb-B* (right hand side). The blue and red lines are the velocity trends given by equations (4) and (5). The green lines are derived by using the upper Hashin-Shtrikman bounds. For details please refer to the text.

[Figure]

Fig. 9: Simulated intrinsic permeability as a function of porosity. Results for the 8 subvolumes of *Carb-A* (left hand side) and *Carb-B* (right hand side) low-resolution samples. Squares markers display each of the pore-solid configurations.

[Figure]

Fig 10: Effective P-wave and S-wave velocities (red and blue dots, respectively) for simulations based on a two-phase segmentation. The results were obtained for subvolumes of the low-resolution images of *Carb-A* (left hand side) and of *Carb-B* (right hand side). The blue and red lines are the velocity trends given by equations (4) and (5); the blue dashed-dotted line is the trend given by equation (6). The experimental results (interval from 0 to 30 MPa confining pressure) are illustrated with crosses connected with yellow bars for comparison. The green dots display the results of the multi-phase simulations (only performed for *Carb-B* and for P-waves). For details please refer to the text.

[Figure]

Fig. 11: Left hand side: intrinsic permeability for subvolume 111 of *Carb-A* low resolution. Permeability calculated from flow simulated in X, Y and Z directions. Right hand side: P-wave velocities for all propagation directions of the same sup-sample. The solid line is the velocity trend given by equation (4). The moderate anisotropy visible in permeability is not present for the velocities.

**Tables**

Table 1: Helium bulk and grain density (in g/cm$^{-3}$), helium porosity (in PU) and air-permeability (in mD) measured at benchtop conditions, for *Carb-A* and *Carb-B*. Permeability of plug *Carb-A* is below the sensitivity level of the apparatus used (0.1 mD). Values are from Vialle et al. (2013).

| Sample ID | Porosity | Bulk density | Grain density | Permeability |
|---|---|---|---|---|
| *Carb-A* | $0.167 \pm 0.002$ | $2.24 \pm 0.01$ | $2.69 \pm 0.01$ | $< 0.1$ |
| *Carb-B* | $0.294 \pm 0.003$ | $1.90 \pm 0.01$ | $2.70 \pm 0.01$ | $60 \pm 5$ |

Table 2: Pressure dependence of the elastic-wave velocities for the two studied samples. Pressure is in megapascal, and P- and S-wave velocity are in kilometers per second. Values are from Vialle et al. (2013).

| Pressure | Carb-A | | Carb-B | |
|---|---|---|---|---|
| | Vp | Vs | Vp | Vs |
| 0 | 5.007 | 2.835 | 2.769 | 1.754 |
| 2.5 | 5.045 | 2.838 | 3.295 | 2.032 |
| 7.5 | 5.091 | 2.835 | 3.477 | 2.191 |
| 10 | 5.100 | 2.834 | 3.504 | 2.226 |
| 20 | 5.116 | 2.822 | 3.551 | 2.295 |
| 30 | 5.113 | 2.821 | 3.553 | 2.290 |
| 25 | 5.114 | 2.821 | 3.548 | 2.284 |
| 15 | 5.085 | 2.822 | 3.527 | 2.268 |
| 5 | 5.048 | 2.840 | 3.384 | 2.195 |
| 0 | 5.033 | 2.842 | 2.792 | 1.785 |

Table 3: CT-scanner parameters used for image acquisition of the two carbonate samples. Sample abbreviations are explained in Table 1.

| Pressure | Carb-A | | Carb-B | |
| --- | --- | --- | --- | --- |
| | Low resolution | High resolution | Low resolution | High resolution |
| Voxel size | $(3.4348\ \mu m)^3$ | $(0.6245\ \mu m)^3$ | $(3.4352\ \mu m)^3$ | $(1.1450\ \mu m)^3$ |
| Image size | 1012 * 1012 | 1012 * 1012 | 1013 * 1013 | 1013 * 1013 |
| Acceleration voltage | 60 kV | 60 kV | 60 kV | 60 kV |
| Current | 80 μA | 80 μA | 83 μA | 83 μA |
| Source-to-sample distance | 50.00 mm | 11.10 mm | 12.00 mm | 12.00 mm |
| Detector-to-sample distance | 50.00 mm | 120.00 mm | 12.00 mm | 60.00 mm |
| Exposure time | 20 s | 20 s | 1 s | 6 s |
| Optical magnification | 4X | 4X | 4X | 4X |

---

## Author Comment (AC3) · 7 Jul 2016

O. Lopez (Referee)
ollop@statoil.com

The overall quality of the paper is good but a number of minor revisions are needed to make it clearer and ensure good understanding. This paper presents a novel approach for estimating elastic properties of carbonate rocks combining laboratory measurements, imaging technique and simulations. The technique in itself is not new but the authors approach differs regarding the image segmentation to estimate unresolved porosity or unidentified phases and then the measured porosity is used for estimating rock properties.

Thank you for this general judgement.

As the editor mentioned, I strongly recommend adding some relevant references regarding previous major works regarding DRP and rock physics: Arns et al., (2002, Geophysics Vol.27), Derzhi et al., (2010, SPE 138586), Ringstad et al., (2013, EAGE) etc: : : The authors should compare their results with previous published works as per today DRP results regarding elastic properties are rarely matching experimental data and are often overestimating Vp and Vs.

We added the suggested and further relevant references.

Part of the author's technique is based on image segmentation into different classes. In Page 6, last paragraph should be improved for better understanding. What defines "global thresholds"? Why do you end up with 5 intermediate classes? Please make it more specific.

We modified the mentioned part of the text.

In page 8, first and second paragraph (line 10) should be clearer. It is difficult here to understand the Figure 7 description you made here. How do you end up with a minimum and maximum porosity for each subsample? You mentioned that you selected the "most relevant subsamples" on which criteria is based your choice (porosity only)? A table for both high and low resolution summarizing both calculated and experimental poro/perm will be maybe useful as it is difficult to understand why Figure 7 and 9 are so different.

Thank you for this comment. We modified the text of both paragraphs and also include an explanation how maximum and minimum porosity is defined with a reference to the corresponding discussion in section 5.2. Permeability results of high and low resolution samples should not be directly compared

as domain sizes are quite different (2.4 mm vs. 0.43 mm characteristic length of low and high resolution sample, respectively). We expect that the domain size of the high resolution sample is too small in order to be representative.

In page 10, Line 18 you highlight the scale issue which is well known from previous studies when comparing DRP and experimental study. Have you made any attempts to upscale DRP results to plug scale? Maybe citation of existing study could be necessary here to avoid misleading conclusions.

You are right, however, we do not have scans on the cm scale for our study. We include a reference to Ringstad et al. (2013).

Paragraph 5.1 you state that even with the highest resolution achievable you cannot resolve all the smallest pores which is true. But techniques exist to overrule these limitations as dry and wet imaging as described by Bhattad et al. (2014, SCA-2014-24) for example. This should be mentioned.

Thank you for your suggestion. We do not agree that the suggested technique will overrule these limitations. However, it is a possible technique to get useful approximations for practical use. Therefore we included the reference.

Paragraph 5.2 page 12, you write that "porosity values of carbonate using micro-XRCT will only provide estimates with relatively high uncertainty due to significant amount of unresolved pore feature in images". I do disagree with this statement; your approach based on single scale imaging is not suitable for proper porosity estimation. Numbers of published papers show the opposite (Lopez et al, 2012). You should be more specific here and mentioned that for carbonate having one image at a single resolution is not enough for porosity estimation.

Thank you very much for your comment. We agree that having one image at a single resolution is not enough for sufficient porosity estimation, but we also disagree about the suitability of the alternative approach presented by Lopez et al. (2012). In Lopez et al. (2012) it is assumed that "the Grey value is a function of porosity below the resolution of the image in the region of the grey value histogram between pore and solid voxels". Since the nature of this function is not further specified in Lopez et al. (2012) we resign any assumptions on such a relationship in this paper and split the grey values between pores and solid voxels into five intervals without any explicit porosity estimate (the only underlying assumption is that lighter grey values correspond to a lower porosity). We modified Section 3.2 according to this comment.

And this is what you work is supporting, that with a single image and doing some assumptions due to unresolved structures it is still possible to estimate some of the effective properties!

Thank you, we will point this out in more detail.

Paragraph 5.4, for the Vp and Vs it would have been nice to have the value at infinite resolution as described by Arns et al.(2002) in their Figure 4c.

Arns et al (2002) derive their "infinite resolutions" considering models with 120^3, 160^3, and 240^3 elements. Our approach is based on a model size of 400^3 and is already benchmarked in Andrä et al (2013). From our point of view the suggested approximation used in Arns et al (2002) and originally derived by Roberts and Garboczi (2000) should not be used in our case.

In summary, this paper demonstrates a new way of estimating elastic properties of carbonates containing micritic phases based on micro-XCRT and experimental nano-indentation. This is an elegant way to define moduli that are often not well known for non-pure minerals and use them for elastic properties determination.

Thanks for this evaluation.

The approach of using X-ray micro-CT in carbonates to obtain P- and V-waves is relative
new and the idea of selecting distinct threshold values for the frontier between
pores and solid matrix is interesting and relevant for the 3D image analysis, presenting
a potential addition to the literature dealing with carbonate and digital rock physics
(DRP).

Thank you for this general evaluation.

However, the paper is somewhat written in a confuse way and should be clarified
and sharpened throughout the text to improve the understandability of the results.

We have considered all relevant comments to address this issue.

The authors should concentrate more in explaining and discussing their results, instead
a big part of the paper deals with the results of other authors with named Tables/Figures
(See e.g.: Page 3, line 15; Page 7, line 4; Page 13, line 5) which is inappropriate since
results from the literature that the authors refer to should be included in the text of
somehow in the paper' structure.

Page 3, line 15: Modified
Page 7, line 4: Modified
Page 13, line 5: Not modified; we regard this accuracy statement with an appropriate reference as
adequate.

There is previous work by all co-authors. However, we are confident that the manuscript would not
benefit from including much more details from these studies. Instead, the interested reader is invited to
consult the references provided in the text. We hope we found a good compromise with the
modifications listed above.

Even though it is important, there was a lack of connection
between the literature and the authors' own results: what was new from the
author's paper compared with the previous literature?

The whole manuscript has been revised carefully with respect to this comment. Not only the abstract
should now be much clearer.

The discussion and conclusions were not very clear; for both sections there is a need of pointing out
and correlating the values and results from the tables and figures in the text.

From our point of view the discussion and conclusions are now clearer. We considered all comments
by the reviewer.

In addition, the theory of P- and S-wave velocities applied to the carbonate samples characterization
should be elucidated. I suggest a previous definition, its importance related to DRP and the analyzed
samples; provide a more detailed discussion between the experimental results and its practical
applications.

We included an explanation how we derive P- and S-wave velocities by our numerical simulations in section 4.1.2. based on a standard workflow already applied in Saenger et al (2004,2011,2016).

Several comments have been incorporated along the text
(see below). Please take them into account (but not limited to) as much as possible.

Thanks for all the suggestions.

The title is too general; The Authors should rename the work to better show the focus of their studies.

Thank you very much for this comment. However, in particular after considering all statements from the reviewers we are even more certain that the title is appropriate.

**Abstract:**
The abstract could be less general also some results (values) should be listed.

Modified

Page 1, line 15: Please list numbers to the resolutions;

Now we specify the range.

 line 17: Mention briefly the properties complemented with nano-indentation;

We include a short statement.

line 20: By "intermediate phases" do you mean "intermediate threshold values for distinct phases"?

Thanks for the hint. Changed.

Lines 21-22: This structure is very confusing. To clarify I suggest the authors giving names to the technique/method used in the laboratory to measure porosity, to the predicted effective properties and to the technique used to acquire the experimental data;

We modified the corresponding sentence.

line 23: Specify that "some sub-samples" actually refers to the distinct smaller regions of interest (ROIs) selected from the acquired CT-datasets. I would also replace "in our case" to "analyzed rocks".

Modified.

**Text: Page 2,**

lines 7-9: When performing 3D images analysis a helpful tool to investigate
and verify the representative elementary volume (REV) of subsamples is
using autocorrelation function. Did the authors investigate REV of their samples somehow?

We modified the corresponding text and added some relevant references. However, from our point of view a detailed study of this aspect is out of the scope of this paper. Also, an auto-correlation function by its own will not allow verifying the REV.

The related literature can help: Haussener, S.; Coray, P.; Lipiᴏnski, W.; Wyss, P.;
Steinfeld, A. Tomography-based heat and mass transfer characterization of reticulate

porous ceramics for high-temperature processing. ASME J. Heat Transf. 2010, 132, 023305:1–023305:9.

Petrasch, J.; Wyss, P.; Stämpfli, R.; Steinfeld, A. Tomographybased multiscale analyses of the 3D geometrical morphology of reticulated porous ceramics. J. Am. Ceram. Soc. 2008, 91, 2659–2665.

Haussener , S.; Steinfeld, A. Effective Heat and Mass Transport Properties of Anisotropic Porous Ceria for Solar Ther mochemical Fuel Generation. Materials 2012, 5, 192–209.

Costanza-Robinson, M.S.; Estabrook, B.D.; Fouhey, D.F. Resentative elementary volume estimation for porosity, moisture saturation, and air-water interfacial areas in unsaturated porous media: Data quality implications. Water Resources Research, 47, WO7513:1–WO7513:12.

Bear,J. Dynamics of Fluids in Porous Media, General Publishing Company LTD, 1972. pp. 19–21.

**Page 2**,
lines 11-13: In which type of material/rock? This statement can be invalid e.g., when analyzing other rock types such as shale;

We added two references with discussed examples and slightly modified our wording.

line 16: "3D rock models"? Maybe, "3D rock pore networks".

We modified the wording to "3D rock pore structure models"

Line 19: It is not the porosity which is smaller, but the pore sizes;

You are right. Thanks for the hint.

lines 20-22: Once more authors draw a statement which is in fact strongly depending on the material/rock type and acquired voxel resolution. Please add rock type and resolution range to correct sentence;

We added a rock type as example and a corresponding reference.

line 28: rephrase sentence.

Rephrased.

**Page 3,**
line 2: Take out "as well";

Ok.

 lines 3-6: How did the authors managed to improve "digital rock images themselves and/or the computational workflow"? Describe it succinctly relating e.g., image enhancement with image acquisition parameters, voxel resolutions, pre- and post-processing;

This is the introduction of the paper. We describe our findings in detail in the following parts of the paper.

line 4: Correct the verb form;

Corrected.

line 6: name the "suggested techniques";

This is the introduction of the paper. We describe our findings in detail in the following parts of the paper.

line 7: Complementary in which aspects? Authors should use
this structure to point out in more details the importance of their work and in which
aspects it is novel and relevant compared with the former cited studies.

Thank you for your suggestion. In contrast to other studies our digital rock physics study is complemented with a very detailed experimental characterization (section 2). Our suggested segmentation technique (section 3) is used to estimate effective mechanical as well as effective transport properties (section 4). Among others, we observe for mono-mineralic (calcite) carbonates a two-phase trend which can be regarded as an upper bound for velocities at all scales (see discussion in section 5) due to the observed self-similarity of those rocks (Jouini et al. 2015).

Pages 3-4, lines 26-7 and Tabs. 1 and 2: Remember using S.I. standard units and note
that the numerical value precedes unit and a space is always used (except for degree,
minute, and second for plane angle) to separate them.

Corrected. Thanks for the hint.

Subtitles are too short and should be improved making a least description of each
subsection.

After carefully considering your suggestion we find our subtitles to be clear with respect to the contents. Also, from our point of view subtitles should not be too long.

Page 4, line 14: "(RMS values)" should be moved to right after "1.4 _m".

Corrected.

Page 5, lines 12-16: Authors mentioned Poisson's ratio but do not say which values
they used for their calculation?

Now we include an additional sentence: "This local Poisson's ratio cannot though be measured experimentally and we have taken here a constant value of 0.3".

Fig. 1 is under explained, e.g., the blue and green
areas mentioned in the caption should be clarified in the text;

From our point of view the explanation in the caption is sufficient.

Explain also which is the relation/implication between the "blue and green"
areas and the nano-indentation results.

See section 5.1: Discussion on these results

Clarify the real mean/relevance of Fig. 1 to the paper context as well.

The "real meaning of Fig. 1" is elaborated in section 5.1. At this point of view we present the laboratory characterization by nano-indentation only.

**Page 6,**
line 1: Inform the source-to-sample distances in Table 3 and change "pixel size"
to "voxel size" adding the cubic unit to the values as well;

Modified.

line9: "illuminate"?

Modified to "remove".

Line 12:
I suggest the authors take out Fig. 2 and only present this sequence in the text itself,
since this workflow is relatively simple and brings no novel information to the paper;

Thank you for your suggestion. We understand that for a reader well versed in this métier this figure
may appear somewhat simple, however we prefer to not take out Figure 2. The simplified visualization
of the workflow applied in this study is supposed to (1) inform the quick reader about the applied
method (in contrast to a rather long description in the text) and (2) to break down the full workflow
into simple single steps to make the method more accessible to readers from other disciplines.

Line 13: Give the voxel size of selected ROI.

This part of the text describes the general workflow; the voxel size for the given ROI can be found in
other parts of the manuscript.

Fig. 4: Authors should describe the dark green areas, which are overlapping volumes between
neighbors subsample ROIs, to improve understanding of their procedure;

Now added in the caption of Figure 4.

 line 18: Keep a standard on typing: "subvolumes" or
"sub-volumes", "subsamples" or "sub-samples";

Modified to "subvolumes" in the whole manuscript.

Lines 20-22: "appropriate dimensions and kernel window sizes" which were?

We modified the sentence and now give the details of the used algorithm.

**Page 7**,
line 7: Were the same Carb-A and Carb-B samples investigated by Vialle et
al., 2013?

Yes, this is stated in the Abstract and in section 2.1:
"Both samples have been characterized in the laboratory in detail in Vialle et al. (2013)."

In positive case, I suggest the authors to add the values of Hg porosity and
compare it succinctly to the He porosity (shown in Table 1) and distinct CT porosities
obtained from the thresholds levels of micritic phases. This will give an idea of the
optimal threshold value which is surely related to the effective rock properties moreover
discussed in the work.

This is a good and attractive idea to use the Hg-porosity versus radius size data given by MICP to
constrain the optimal pore size threshold from the CT data. However, it is challenging as MICP gives
a pore throat (or access) size and not the actual pore size invaded by mercury. Hence the two "pore
size" versus porosity data express different relationships that are difficult to correlate.

Line 14-15: give the used values for pressure bound condition
and dynamic viscosity of fluid;

In our numerical simulation delta p is -5.8 e-4 Pa/m an µ is 1.2 Pa s. The corresponding text passage has been modified.

line 19: what does the form "RSG" stand for? Note that

Corrected.

Fig. 6 was not commented in the manuscript text. If Fig. 6 isn't that relevant to the paper' findings it should otherwise be taken out.

In paragraph 3.2 we reference Figure 6. Figure 6 is important to illustrate the different segmentation classes.

**Page 8,**
 line 6: What do you mean by "most relevant subsamples"? Give the criteria to judge a subsample relevant;

Corrected. Please also refer to our answer to this point to reviewer #1 (Oliver Lopez).

also do the authors mean by "numerical investigation"
in this structure the P-and S-waves velocities? Please clarify! Because if one looks to
the numerical investigation of permeability (Figs. 7 and 9) it is possible to see that simulations
were performed in all 8 subsamples, while P-and S-waves velocity simulations
are given only for one subsample (give the subsample names in the legend) of each
carbonate (Figs. 8 and 10).

Corrected. We have improved the Figures according to this comment.

In Fig 7,
add "simulated" after "Intrinsic permeability" and
in the graphic axis (in Fig. 9 as well).

Modified.

Comparing the results of permeability simulations
for the high resolution (Fig. 7) and the low ones (Fig. 9) one can see that only the
minimum and maximum threshold values were depicted in Fig. 7. Please elucidate the
reasons for that.

Now we explain the selection in Section 4.1.1.

Lines 11-18: Authors made a good observation and should justify this
result better.

The paragraph is part of a results section. Justification or evaluation of results and observations should
be part of the discussion.

Another interesting find when comparing Figs. 7 and 9 is the variation
on the permeability results between subsamples: for the low resolution results less
variation in the permeability is observed compared to the higher resolution, indicating
less anisotropy of the subsamples and more material representativity. It is an important
find in your study, you have it in numbers and you should highlight it!

As mention in the answer to the question of the first reviewer, we could not directly compare results
shown in Fig. 7 and Fig. 9. The permeability results in Fig. 7 and Fig. 9 do not give any information

about anisotropy. We perform the Stokes flow simulations only in one direction (z-direction, see Fig. 3), which we choose identical to the direction of wave propagation. We have modified section 4.1.1 to clarify this point.

Observe as well
how the subsamples of Carb-B (high resolution) showed to be heterogeneous; even
though as the authors describe "it shows a much lower variation between the extreme
values", the subsamples have extreme variation in the permeability values compared
with subsamples of Carb-A. Which would be the probable causes for these results?

As mentioned also in the discussion, the high resolution samples are probably too small to be representative. The results in permeability calculations for different subvolumes, Fig. 7 left/right, show that fact. In addition, we can also observe that a variation in extreme values of porosity (for Carb-B high resolution, Fig. 7, right) doesn't seem to affect the permeability results. This could also be a result of a non-representative, i.e. too small subvolume/sample (e.g. a disconnected pore dominating the porosity value of the subvolume).

Line 18: This statement is half wrong!

Thanks for the comment. We fully agree because one statement could be misunderstood.
The high resolution samples don't seem to be representative as already mentioned above. Therefore large disconnected pores could dominate the porosity changes without contributing to the effective permeability.

Line 20: Here the "micritic phases" term is given
without a clear explanation that they actually are the distinct phases identified from the
threshold' classes of 3D images (as described in section 3.2). Please clarify it also
linking it to the Fig. 5.

We modified the corresponding text. Thanks for this hint.

The same is happening in section 4.2.1 when a new term "six
possible domains" is introduced.

We have modified the text to clarify this point.

**Page 9,**
 line 12: In fact the threshold values are being varied what implies in the porosity
change!

Indeed the porosity changes. We want to calculate the permeability for the porosity variation of each subsample.

Lines 18-19: Make sure to inform that these results are shown in Fig. 11;

The sentence has been modified.

Lines 22-23: Rephrase structure;

Rephrased.

lines 25-26: Rephrase the position of "(Figure 10)" in the structure.

Rephrased.

**Page 10,**

lines 1-4: I disagree that only Carb-B showed slightly difference, which can also be seen in the P-wave results of Carb-A, on which data "a blue dashed-dotted line" should be fitted as well.

Ok; agreed. Now we include also "a blue dashed-dotted line" for Carb-A.

IMPORTANT: Note that if P-waves are represented with the blue color in Figs 8 and 10, captions must be corrected.

All captions are corrected according to this comment.

Lines 13-16: The performed procedure and described results are very interesting for a better discussion;

We added some sentences to explain the procedure in a better way.

lines 21-23: Make a link to it commenting the finds from Carb-B (Fig.7) and discussing in a practical manner how the present work overcomes this problem.

We modified the corresponding lines: "In general, a multi-scale approach as suggested by Ringstad et al. (2013) should be used for upscaling the results to the plug scale. However, our studies on Carb-A and Carb-B will suggest workflows which should be applied in practice for as many samples as possible for improving the statistical significance."

**Page 11:**
Section 5.1: Although the idea of correlating estimated elastic properties of carbonates based on distinct micritic phases identified from the threshold' levels in micro-XCRT images, with experimental nano-indentation experiments sounds very attractive, the authors failed in their expectations described in the lines 22-24.

We are not exactly sure why the reviewer points to some expectations we described in the lines 22-24. From our point of view we demonstrate and describe clearly that the results from indentation experiments are difficult to use in a direct way. However, as explained in the text, the results motivate us to use effective medium approaches at the image scale.

For a rock/material having a defined amount of pores and solid matrix, one can expect an image threshold with at least two peaks: one in the darker gray levels regions (related to the pores) and another in the brighter regions (linked to the matrix);

This is true. However, as shown in Figure 5, bottom row, right hand side, there are also examples where the peak corresponding to pores will not show up as clear as for other examples.

however If the analyzed material has also a certain amount of heavy phases (e.g. iron) then another additional peak in the threshold can be observed.

In this paper we only consider mono-mineralic carbonates (see Section 2.1).

Whereas (as the authors described very well) it is difficult to see the moduli peaks of pores in the nano-indentation experiment results, naturally because the values are very low.

Ok, thanks.

The relation from the micro-XRCT images and nano-indentation experiments using the number of threshold peaks seems somehow inappropriate.

In the manuscript we clearly state that: "Therefore the direct translation of moduli derived fom nanoindentation remains also to be difficult". As mentioned above, the results from nano-indentation will suggest that the use of effective medium approaches on the considered image scales is appropriate.

**Page 12,**
lines 5-8: Include figures numbers (low and/or high resolutions) of your work
to improve reading and understanding;

Now included. Thanks for the hint.

line 16: nane the technique used to the measured
porosity or add "as shown in Table 1".

Modified.

In line 18: specify "full sample".

Specified.

Lines 22-24: Authors should be careful and add in this statement, that this observation is for their specific case (Carb-A and Carb-B) within the investigated resolutions which is based on the single image scales. Nowadays the use of multi-scale approaches to investigate porosity and DRP of heterogeneous rocks such as carbonates became widely common and has proving to be reliable.

We disagree with this statement. As stated in the paper we conclude that the porosity values of carbonates using micro-XRCT-data will only provide estimates with a relatively high uncertainty due to the significant amount of unresolved pore features in the images. The reviewer is kindly asked to give a reference where porosity has been determined successfully based on micro-XRCT-data **and** the procedure has been described in detail.

**Page 13**,
lines 2-3: name the tables/and figures from were readers can see these results;

Thanks for the comment. We have added the respective figure references in the text.

line 9: change to "experimental measurement".

Modified

Line 12: name the porous materials;

Changed to "porous rocks"

lines 15-16: How "statistically significant" (also given the Summary) samples should be? Try to base it on your results with the proposed approach using multi micritic phases and subsamples (ROIs).

The number of samples necessary to be "statistically significant" is chosen depending on the domain size. In case of the numerical simulation we have chosen 8 samples for the permeability calculations. We have clarified this point at the end of section 5.3.

**Page 14,**
section 5.5: the statement that "any significant anisotropy for permeability" was found in the analyzed samples is in disagreement with some of the paper' results (see e.g. Fig. 7). Elucidate the anisotropy changing from the higher to the lower

There are no anisotropy findings in Figure 7; our results on anisotropy are displayed in Figure 11. As explained in section 5.5 only a few samples are out of this trend which is displayed in Figure 7. This moderate anisotropy is regarded as not significant.

Concerning to the Summary:
Summary is in general written in a confuse way making
it hard to follow the author's thoughts. The summary should be rewritten in a more focused and brief way. Again, the authors provide their conclusions without backing them up with the quantified values that they base their assumptions on, making the work appear somewhat subjective. They tend to loose themselves in generalizations such as "the porosity of the rock samples is the most relevant parameter"; certainly the authors do not mean that for any purpose in the world including rocks porosity is the most relevant parameter, as an example for structures that need to be sharpened and detailed.

The Summary has been revised substantially according to this helpful comment.

Several references are missing, i.a.:

Page 2,
lines: 10-11,

Fusseis et al., 2014

14-15,

Andrä et al. 2013b

15-18,

Osorno et al., 2015, Saenger et al., 2016

22-24;

Andrä et al. 2013b

Page4, line 22;

Fischer-Cripps (2004), Lebedev et al. (2014)

Page 5, lines 3,

Fischer-Cripps (2004)

Page5, line 7;

Fischer-Cripps (2004).

Page 6, line 8 (reference the model used in the reconstruction);

We have given in the text the name of the software we used but have slightly modified the sentence to make it clearer.

Page 6, line 20;

From our point of view the reference "FEI Visualization Sciences Group" is sufficient.

Page 7, lines 7, 10;

The reference was given in line 9 "Osorno et al., 2015"

Page 13, line 27.

No reference required. We modified the sentence accordingly.